# Rapid estimation of seismic intensities by analysing early aftershock sequences using the robust locally weighted regression program (Lowess)

Huaiqun Zhao [1], Wenkai Chen [1], Can Zhang [2], Dengjie Kang[1]

[1] Lanzhou Institute of Seismology, China Earthquake Administration, Lanzhou 730000, China

[2] Sichuan Earthquake Administration, Chengdu 610041, China

*Correspondence to*: Wenkai Chen (cwk2000@yeah.net)

**Abstract.** Accurate and rapid assessment of seismic intensity after a destructive earthquake is essential for efficient early emergency response. We proposed an improved method, AL-SM99, to assess seismic intensity by analysing aftershock sequences that occur within 2 hours of mainshocks. The implementation effect and application conditions of this method were illustrated using 27 earthquakes with Mw 6.5–8.3 that occurred globally between 2000 and 2023. When the fault system in the seismic region is clear and simple, the robust locally weighted regression program (Lowess)-fitted curves could be used to estimate the location and length of the fault rupture. Lowess results can indicate the overall rupture trend and make reliable rupture scale judgments even when the fault system is complex. When Mw $\geq$ 7.0 and the number of aftershocks exceeds 40, the AL-SM99 intensity evaluation results may be more reliable. Using aftershock catalogues obtained by conventional means allows for a stable assessment of seismic intensities within 1.5 hours of the mainshock. When the number of aftershocks is sufficiently large, the intensity assessment time can be greatly reduced. With early accessible aftershocks, we can quickly determine the rupture fault planes and have a better estimate of the seismic intensities. The results of the intensity assessment provide a useful guide for determining the extent of the hardest-hit areas. By expanding the data sources for seismic intensity assessment, the early accessible data are utilised adequately. This study provides a valuable reference point for investigating the relationship between early aftershock events and fault rupture.

## 1 Introduction

Seismic intensity reflects the strength of ground motion caused by an earthquake and its influence at a certain location. Rapid and accurate assessment of seismic intensity facilitates the development of emergency measures in the aftermath of a destructive earthquake, thereby reducing the number of

fatalities and property damage (Erdik et al., 2011; Poggi et al., 2021). Therefore, it is necessary to develop methods for the rapid assessment of seismic intensity and the effective utilization of disaster data in the early post-earthquake period.

Various methods have been employed to evaluate seismic intensities. In regions, such as Japan, with a dense distribution of seismic monitoring stations, the density of intensity meters and stations is sufficient

to support the assessment of intensities (Nishimae, 2004). The ShakeMap system of the US Geological Survey (USGS) combines predicted ground motion values with station observations to determine the seismic intensity of a region and publishes the results online in near real-time (Worden et al., 2020). Empirical models, particularly elliptical attenuation models, are widely utilised in China, where the relevant authorities conduct on-site investigations during a certain period after an earthquake to draw and

publish macroseismic intensity maps (Wang et al., 2013; Xu et al., 2020). Furthermore, deep learning-based real-time seismic intensity prediction has emerged as a current research focus (Otake et al., 2020; Chen et al., 2022). Internet data, remote sensing, and radar data are widely utilised in earthquake damage assessment (Dell'acqua and Gamba, 2012; Hao et al., 2012; Xu et al., 2013; Yao et al., 2021).

The time between the occurrence of an earthquake and the first acquisition of disaster data from the

seismogenic region, typically within 2–3 hours of the mainshock, is defined as the black box period for earthquake emergency response (Nie and An, 2013). In general, little information regarding a disaster has been reported from the disaster area during this period, despite the fact that decision-makers require reasonably reliable data to initiate emergency response efforts. Seismic monitoring networks and the internet are the main sources of data (Nie et al., 2012; Xia et al., 2019). Consequently, ground-motion

assessment maps are typically generated using data from dense intensity meters, empirical models, or internet-based seismic intensity assessment systems, and they serve as the foundation for early emergency command and decision-making (Wald et al., 1999; Atkinson and Wald, 2007; Sokolov et al., 2010). Back projection could image the fault geometry of large earthquakes at high resolution and is frequently used to trace surface rupture processes and source durations (Ishii et al., 2005; Wan et al.,

2022). The combination of back-projection results and P-wave amplitudes could be used to quickly estimate the source length and magnitude of large earthquakes (Wang et al., 2017). Using the back-

projection technique and ground motion prediction equation (GMPE), Chen et al. (2022a) developed a new algorithm for quickly obtaining the intensity maps of destructive earthquakes. The algorithm was validated during the emergency response phase of the 2021 Maduo Mw 7.3 and 2021 Yangbi Mw 6.1

earthquakes in China and was confirmed to be suitable for intensity assessment in regions with sparse observation networks (Chen et al., 2022b).

The spatial distribution of aftershock sequences after large earthquakes reflects surface rupture information. Aftershock sequences are widely utilised in studies to investigate the structure and nature of causative faults and the process of earthquake nucleation (Umino et al., 2002; Bachura and Fischer,

2016). Artificial intelligence (AI) can extract valuable information and patterns from massive amounts of data, and it is frequently used in seismology to improve phase detection sensitivity while processing massive amounts of real-time monitoring data (Jiao and Alavi, 2020). The use of machine learning enables more sensitive identification of shake events and increases the number of detected earthquakes compared to routine methods (Liu et al., 2020). Relocated aftershock sequences have become one of the

most important tools for studying the rapid determination of causative faults after an earthquake (Fuis et al., 2003; Wang et al., 2021). The spatial distribution of aftershocks may reflect the continuation of sliding at the edge of the area of maximum co-seismic displacement or the activation of subsidiary faults at the rupture boundary of the mainshock (Mendoza and Hartzell, 1988), dynamic stresses could remotely trigger seismic activity and may have the same effect in near-fault regions (Kilb et al., 2000). The early

aftershocks of the 2008 Wenchuan Mw 7.9 earthquake were located below or around the mainshock slip patch boundaries, and the geometry of the fault limited the occurrence of early aftershocks within the first 24 hours (Yin et al., 2018). Aftershock sequences that occur shortly after the mainshock could outline the basic characteristics of the mainshock rupture surface (Kisslinger, 1996). The area of the aftershock zone is a good first-order approximation of the mainshock rupture area as the aftershocks tend

to concentrate near the boundary of the mainshock rupture (Neo et al., 2021). mainshock and aftershock sequence simulations in geometrically complex fault zones have shown that early aftershocks are good indicators of the extent of mainshock rupture; it is reasonable to estimate the length of the fault plane based on well-constrained aftershock locations, and most aftershocks are distributed within 1–1.5 km of the mainshock rupture (Yukutake and Iio, 2017; Yabe and Ide, 2018; Ozawa and Ando, 2021).

As mentioned previously, it is possible to extract fault rupture information from early aftershock data. The seismic intensities assessed by Zhao et al. (2022b) using aftershocks within two hours of the 2022

Menyuan Mw 6.6 earthquake and GMPE based on fault rupture agreed with the on-site investigation results. Although the seismic intensity assessed by this method was useful for identifying the hardest-hit areas, no in-depth analysis of the selection and application of aftershock data was conducted. In this study, we propose a method for rapidly assessing the seismic intensity following an earthquake by analysing the spatial distribution of aftershock sequences using the Lowess. The interquartile range (IQR) was utilized to exclude aftershocks with abnormal geographic coordinates from the aftershock sequence that occurred within 2 hours of the mainshock. The geographic coordinates of the processed aftershocks are then fitted with Lowess, and the results of the fitting are used in the GMPE calculation. Finally, the ground motion calculation results are converted to seismic intensity using the seismic intensity scale. The implementation of the new method and the effect of intensity assessment are demonstrated for specific earthquake cases, and its applicability is discussed.

## 2 Data and methods

### 2.1 Data

The earthquakes chosen for this study had a magnitude (Mw) greater than or equal to 6.5 and were shallow (hypocentre depth less than 70 km). When the Modified Mercalli Intensity (MMI) is VII, ShakeMap uses the terms "very strong" and "moderate damage" to describe the levels of impact on a region (Worden et al., 2020). Similar descriptions of intensity VII exist in the Chinese Seismic Intensity (CSI) scale. For the intensity range of VII–VIII, human perception of shaking began to saturate, and it may be difficult to distinguish seismic intensities above VII based on the individual descriptions of the felt shaking alone. (Dengler and Dewey, 1998; Worden et al., 2020). To ensure that the cases included in the study caused significant effects within the seismogenic region and conveniently compare our results with those assessed by ShakeMap, it was specified that the maximum intensity of the ShakeMap evaluation for the cases used in this study could not be less than VII. The more aftershocks that occur within a 2-hour period, the more probable that the length, direction, and other aspects of the surface rupture are outlined. The events chose in this study with a minimum of 15 aftershocks within 2 hours after the mainshock. Twenty-seven earthquakes between 2000 and 2023 that met the aforementioned criteria were chosen, and their seismic station data and isoseismal lines were downloaded in shapefile format from ShakeMap (https://earthquake.usgs.gov/data/shakemap/).

Aftershock sequences for all earthquakes were downloaded from the International Seismological Centre (ISC), except for those that occurred in China. As the ISC only recorded a small number of the aftershocks that occur within 2 hours of Chinese earthquakes, we chose to download aftershock data from the earthquake catalogue repository of the Earthquake Science Knowledge Service System in China. For the earthquakes that occurred in China, we digitised the intensity map drawn by the on-site investigation

published by the China Earthquake Administration (CEA) using ArcGIS. The global Vs30 was downloaded from the US Geological Survey website, and the earthquake occurrence time used was Universal Time Coordinated.

## 2.2 Methods

### 2.2.1 Outlier selection and deletion

In the raw aftershock sequence data, there may be isolated aftershocks far from the aftershock cluster area. These aftershocks are regarded as outliers, affecting the judgement of the distribution trend of aftershocks and causing the Lowess result to deviate. IQR is a straightforward method in descriptive statistical data analysis that is frequently employed to identify outliers in a variety of fields (Rojas-Martinez et al., 2007; Spitzer et al., 2014). It is expressed as the difference between the third (Q3) and

first (Q1) quartiles, i.e. IQR = Q3-Q1. The quartile range [Q1-kIQR, Q3+kIQR] is utilized as the criterion for identifying outliers (Perez and Tah, 2020). If the aftershock longitude or latitude value fell outside this range, it was deemed an outlier and deleted; the default value for k in R software is 1.5. In this study, outliers were selected and eliminated using the R programming language.

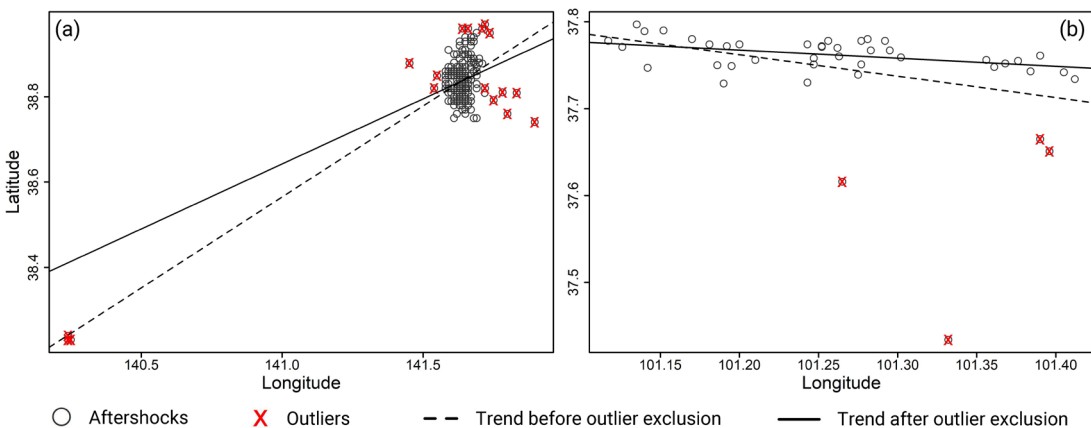

**Figure 1:** Aftershock sequence spatial distribution before and after deleting outliers. The (a) 2003 Miyagi-Oki Mw 7.0 and (b) 2022 Menyuan Mw 6.6 earthquakes.

If the raw data contain outliers, the fitting line will be biased toward sporadic aftershocks. After removing these events, the new trend line is more consistent with the spatial distribution of the aftershock sequence (Fig. 1). The preserved aftershocks contain useful information concerning surface rupture. If there are no outliers in the original data, it can be used directly for Lowess. IQR checking revealed no outliers in the aftershocks that occurred within 2 hours of the 2008 Wenchuan Mw 7.9 and 2016 Kaikōura Mw 7.8 earthquakes (Table 1).

Table 1: Number of aftershocks and identified outliers for the selected earthquakes.

| | Data | Location | Magnitude | Aftershocks | Outliers |
|---|---|---|---|---|---|
| 1 | 20001006 | Matsue (Japan) | 6.7 | 152 | 4 |
| 2 | 20030526 | Miyagi-Oki (Japan) | 7.0 | 259 | 24 |
| 3 | 20051008 | Kashmir (Pakistan) | 7.6 | 54 | 0 |
| 4 | 20080512 | Wenchuan (China) | 7.9 | 43 | 0 |
| 5 | 20080613 | Iwate-Miyagi Nairiku (Japan) | 6.9 | 227 | 1 |
| 6 | 20100404 | Baja California (Mexico) | 7.2 | 60 | 2 |
| 7 | 20100903 | Darfield (New Zealand) | 7.0 | 139 | 2 |
| 8 | 20110411 | Hamadoori (Japan) | 6.6 | 79 | 12 |
| 9 | 20111023 | Van (Turkey) | 7.1 | 46 | 6 |
| 10 | 20130816 | Grassmere (New Zealand) | 6.5 | 46 | 3 |
| 11 | 20150425 | Gorkha (Nepal) | 7.8 | 68 | 14 |
| 12 | 20150916 | Illapel (Chile) | 8.3 | 56 | 4 |
| 13 | 20160415 | Kumamoto (Japan) | 7.0 | 538 | 0 |
| 14 | 20161030 | Preci (Italy) | 6.6 | 89 | 6 |
| 15 | 20161113 | Kaikōura (New Zealand) | 7.8 | 106 | 0 |
| 16 | 20171112 | Sarpol-e Zahab (Iraq) | 7.4 | 15 | 2 |
| 17 | 20180504 | Hawaii (America) | 6.9 | 38 | 1 |
| 18 | 20180905 | Tomakomai (Japan) | 6.6 | 162 | 6 |
| 19 | 20180928 | Palu (Indonesia) | 7.2 | 18 | 2 |
| 20 | 20181130 | Anchorage (America) | 7.1 | 127 | 9 |
| 21 | 20190706 | Ridgecrest (America) | 7.0 | 105 | 2 |
| 22 | 20201030 | Samos (Greece) | 7.0 | 97 | 10 |
| 23 | 20210521 | Maduo (China) | 7.3 | 70 | 1 |
| 24 | 20220107 | Menyuan (China) | 6.6 | 43 | 4 |
| 25 | 20220905 | Luding (China) | 6.6 | 78 | 8 |
| 26 | 20230206 | Pazarcik (Turkry) | 7.8 | 27 | 5 |
| 27 | 20230206 | Elbistan (Turkey) | 7.5 | 24 | 0 |

**2.2.2 Application of Lowess**

Fitting methods are frequently used to visualize trends in scatterplots and predict the future development

of the research object. Since the introduction of local fitting for processing time series data into the

general situation of regression analysis, local regression methods have developed continuously

(Macaulay, 1931; Watson, 1964; Stone, 1977). Cleveland (1979, 1981) proposed a single variable local

smoothing method, a robust locally weighted expression, and a programme (Lowess) for using this

method to smooth scatter plots. Later, a multivariate smoothing model called Loess was also introduced

(Cleveland and Devlin, 1988). Lowess has been utilized to visualize trends in scatterplot distributions in

a variety of natural and social sciences (Quackenbush, 2002; Tetlock, 2007; Grimmer and Stewart, 2013;

Law et al., 2014). Lowess improves visualization of geophysical and high-frequency financial data trends

and is simple to implement (Mariani and Basu, 2014).

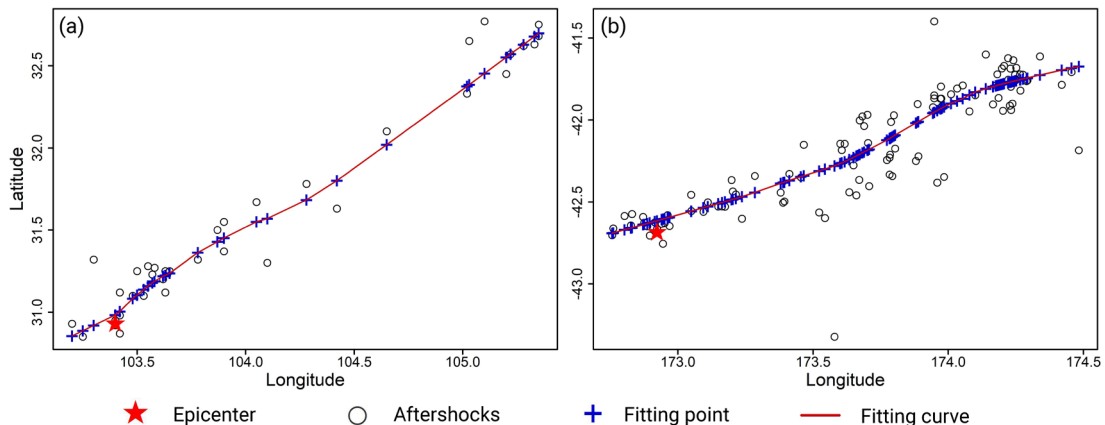


**Figure 2:** Lowess fitting results for aftershocks that occurred within 2 hours of the (a) 2008 Wenchuan Mw 7.9 and (b) 2016 Kaikōura Mw 7.8 earthquakes.

Figure 2 shows the Lowess fitting results for aftershocks that occurred within 2 hours of the 2008

Wenchuan Mw 7.9 and the 2016 Kaikōura Mw 7.8 earthquakes. The early aftershocks of these two events

were distributed along the surface rupture zone and within a certain range on both sides of the zone,

based on the spatiotemporal distribution of the aftershock sequences (Fig. 3). The early aftershocks of

the Wenchuan earthquake occurred within 300 km of the epicentre and were concentrated along the main

rupture direction, and those of the Kaikōura earthquake were distributed within 200 km of the epicentre

and relatively dispersed along the main rupture strike, which was primarily caused by the complex fault

system (Wallace et al., 2018). Early aftershocks in both cases ruptured in a single direction. Based on

this, we believe that the fitting results in Fig. 2 can depict the length and direction of the earthquake

rupture.

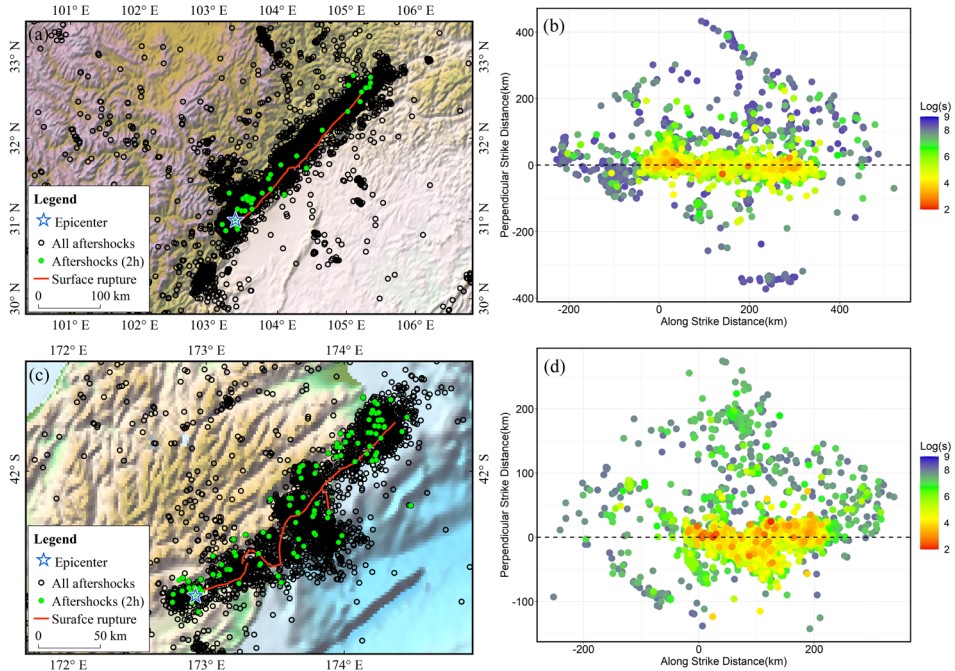

**Figure 3:** Spatiotemporal distribution of early aftershock sequences after the mainshock. (a) and (c) Spatial distribution of aftershocks of 2008 Wenchuan Mw7.9 and 2016 Kaikōura Mw 7.8 earthquakes. (b) and (d) Temporal distribution of aftershocks from Wenchuan and Kaikōura earthquakes. The s in Log(s) denotes the time in seconds between the aftershock and mainshock.

### 2.2.3 SM99 GMPE

Si and Midorikawa (1999) obtained the GMPE, which we call SM99, by fitting 21 records of strong shocks that occurred in Japan between 1968 and 1997. This equation exploits the shortest fault distances and calculates ground motion using the geometry of the imaged faults, has been validated by historical examples and actual seismic emergencies, and found to be applicable in western China in addition to Japan (Kamigaichi et al., 2009; Si et al., 2010; Zhao et al., 2022a). Rapid seismic intensity assessment has been accomplished using fault data or back-projection techniques based on this GMPE, providing a powerful guarantee for earthquake emergency response (Zhang et al., 2021; Chen et al., 2022b). In this method, the accurate inversion of the source rupture process and selection of GMPEs with good applicability remain the focus of research. We calculated the ground motion value by substituting the fault data with the Lowess result of aftershock sequences obtained within 2 hours of the mainshocks. The GMPE is not modified, and equations can be referred to in the relevant literature (Si and Midorikawa, 1999; Chen et al., 2022).

As previously mentioned, the improved method consists of three essential components: Aftershocks, Lowess, and SM99 GMPE. It rapidly assesses the seismic intensity using two main steps: fitting the trend

of spatial distribution of aftershocks using Lowess and then calculating the ground motion utilizing SM99 GMPE. We named the method AL-SM99.

## 3 Results

### 3.1 AL-SM99 application results

#### 3.1.1 The 2008 Wenchuan Mw 7.9 earthquake in China

The 2008 Wenchuan Mw 7.9 earthquake was one of the most devastating seismic events in China in recent years (Chen et al., 2018). AL-SM99 was used to calculate the peak ground velocity (PGV) and site-corrected PGV values (we called $PGVvs_{30}$) for this earthquake. The ground motion value was then converted to seismic intensity using CSI and the ground-motion–intensity conversion equation (GMICE, Worden et al, 2012; 2020), respectively.

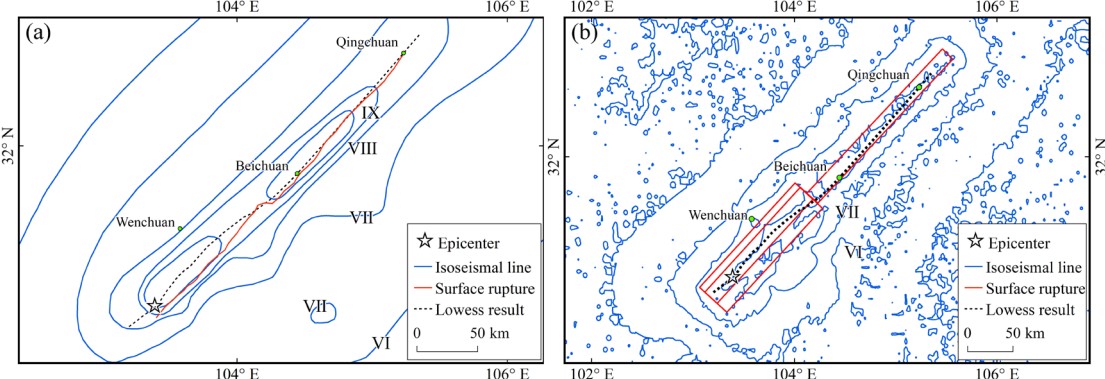

**Figure 4:** Spatial distribution of the 2008 Wenchuan Mw 7.9 earthquake aftershocks fitted using Lowess. Comparison with (a) actual surface rupture and (b) ShakeMap fault planes.

The curve obtained by Lowess smoothing for the aftershocks that occurred within 2 hours of the mainshock exhibited reasonable agreement with the length and direction of the actual surface rupture, particularly from Beichuan to Qingchuan, where it almost coincided with the actual surface rupture and lay entirely within the ShakeMap-published planar distribution of faults (Fig. 4). The seismic intensities evaluated from the raw aftershocks data, whose geographical coordinates were directly utilized in the SM99 calculation, can roughly infer the extent of the hardest-hit area, and the intensity zone boundaries in the direction of the causative fault were found to be agree with both the CEA and ShakeMap intensity maps. The spatial dispersion of aftershocks results in an excessively broad assessment range of intensity VIII when compared to those of ShakeMap and CEA. This was particularly evident between Wenchuan

and Beichuan, where intensity estimates were approximately 1 degree higher than those derived from the ShakeMap. However, the boundaries between intensity regions were extremely consistent with those determined by ShakeMap (Fig. 5).

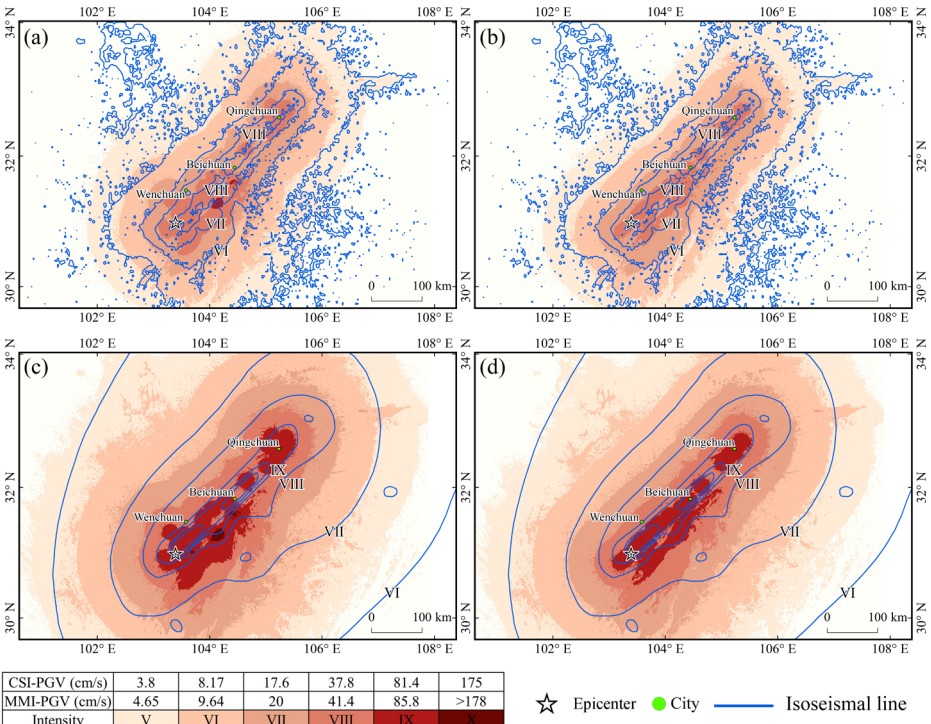

| CSI-PGV (cm/s) | 3.8 | 8.17 | 17.6 | 37.8 | 81.4 | 175 |
|---|---|---|---|---|---|---|
| MMI-PGV (cm/s) | 4.65 | 9.64 | 20 | 41.4 | 85.8 | >178 |
| Intensity | V | VI | VII | VIII | IX | X |

☆ Epicenter    ● City    —— Isoseismal line

**Figure 5:** Assessment of the 2008 Wenchuan Mw 7.9 earthquake intensity. Comparison of seismic intensities from (a) unprocessed aftershocks and (b) Lowess results with those obtained from ShakeMap. Comparison of intensities from (c) unprocessed aftershocks and (d) Lowess results with macro-earthquake intensities obtained from the CEA. The MMI-PGV values in the legend match the values in the ShakeMap instrumental intensity scale legend (Worden et al, 2020).

Figure 6 shows the profiles of different seismic intensity assessments. The maximum MMI values based on the PGVvs$_{30}$ assessment are consistent with the ShakeMap results, with a significant match in the range of intensity zones from Beichuan to Qingchuan. The highest intensity assessed on the CSI scale was X, which was one degree less than the published macro intensity value from the CEA. Both results have roughly the same amount of the intensity in the IX region, and macroseismic intensities X–XI are also situated within this range. In addition, the intensity assessment results from Beichuan to Qingchuan were ~1 degree lower than the CEA results. However, the boundary locations of the various intensity regions are remarkably similar. For example, the boundaries of intensity regions VII and VI determined by our method roughly overlapped those of intensity regions VIII and VII determined by the CEA, respectively. As previously mentioned, it is difficult for people to accurately assess the effects of VII and

higher intensities based solely on perception. That is, the ground motion values calculated using GMPE

are instrumental intensities, reflecting a high level of strong shaking and unaffected by factors such as

population and buildings (Worden et al, 2020). Therefore, we consider this result to be normal. The AL-

SM99 result was highly consistent with published intensity ranges from the CEA for the south-western

region of Beichuan.

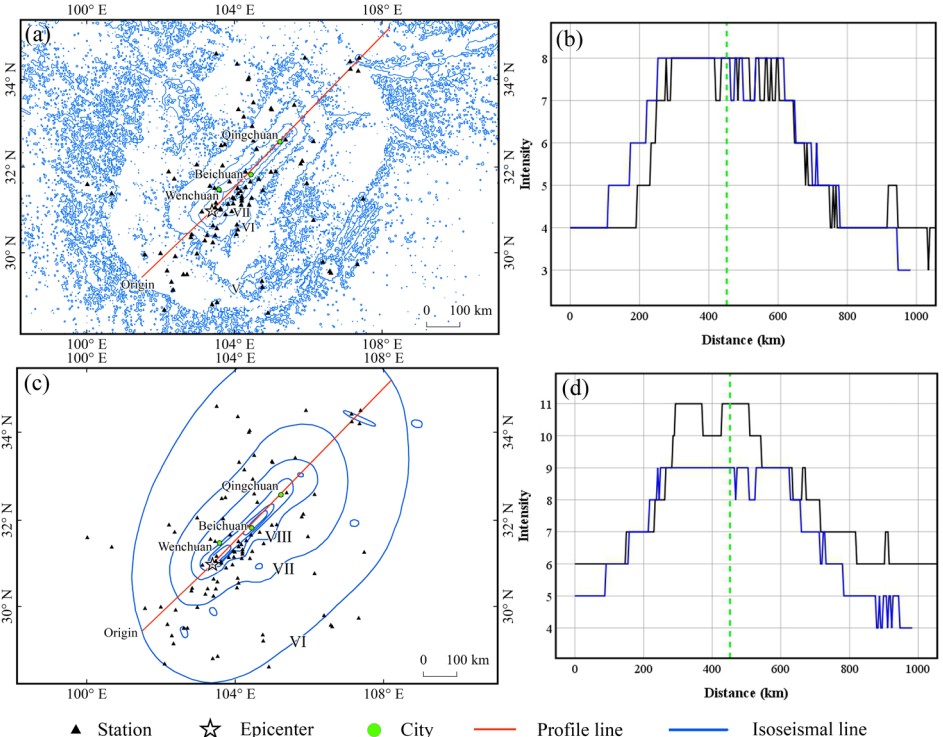

**Figure 6:** Profiles of seismic intensity assessed using Lowess results. Profiles are drawn along the red lines in (a) and (c). (b) Comparison with profiles of seismic intensities obtained from ShakeMap and (d) comparison with profiles of seismic intensities obtained from the CEA. The green dashed line in the profile line is the location of Beichuan, the blue profile line represents the seismic intensity from this study and the black one represents those from ShakeMap or CEA.

Table 2 demonstrates that the $PGVvs_{30}$ values calculated by AL-AM99 were remarkably similar to those

calculated by ShakeMap. Notably, ShakeMap considers station records of high quality when calculating

ground motion, whereas the results of this study were not corrected using station data. The hardest-hit

areas assessed by the Lowess-smoothed data were more convergent and had a clearer extent compared

to the unprocessed aftershock sequences, and the length along the causative fault direction was not

significantly altered. This demonstrates the potential of AL-SM99 for applications where the fault system

in the seismogenic region is straightforward, and the rupture scale of the source is evident. Aftershocks

recorded in the first two hours following an earthquake can be used to portray more accurate information

concerning the rupture direction and length using the Lowess curves. The assessment results are crucial in identifying disaster regions and prioritizing the deployment of rescue forces.

**Table 2:** Effect of AL-SM99 in predicting the PGV of the Wenchuan earthquake within the assessment region for ShakeMap intensity V.

|  | Residual (log10(obs./calc.)) | R² | RMSE | MAE |
|---|---|---|---|---|
| **AL-SM99** | -0.169 | 0.461 | 22.058 | 11.757 |
| **ShakeMap** | 0.039 | 0.453 | 22.225 | 9.523 |

### 3.1.2 The 2016 Kaikōura Mw 7.8 earthquake in New Zealand

On November 13, 2016, a Mw 7.8 earthquake struck the Kaikōura region of New Zealand. It ruptured on the surface of approximately 20 faults and exhibited a highly complex form that exceeded the traditional perception of multi-fault ruptures (Litchfield et al., 2018; Wallace et al., 2018).The relocated aftershock sequence of this earthquake was almost entirely concentrated in the shallow crust of the upper plate, and few aftershocks occurred at the subduction interface (Lanza et al., 2019).

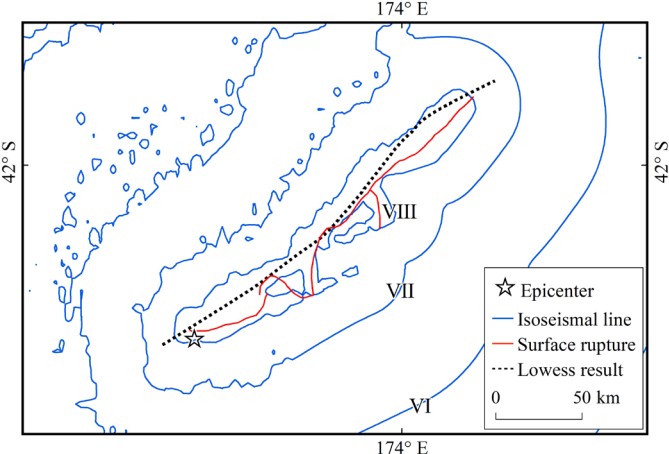

**Figure 7:** Comparison between the Lowess fitting curve and actual surface rupture of the 2016 Kaikōura Mw 7.8 earthquake.

Fitting the aftershock sequence obtained within 2 hours of the mainshock yielded a southwest-northeast curve (Fig. 7), which was consistent with the overall directional distribution of the surface rupture (Kaiser et al., 2017). The length of the curve was slightly longer than the actual surface rupture, and during the early post-earthquake period, the basic pattern of surface rupture could be tentatively determined. The GMICE was used to convert the predicted PGV$vs_{30}$ values to MMI and generate intensity assessment maps (Fig.8). Although no outliers were identified during the pre-processing phase, the range of assessed

earthquake intensities based on the raw aftershock sequence was excessively broad. Nearly the entire estimated area of intensity VIII falls within the range of ShakeMap intensity VII. In addition, individual aftershocks were spatially dispersed but were not deemed to be outliers, resulting in zones of anomalous

intensity (Fig.8(a)). This outcome impedes the accurate determination of areas with varying degrees of damage. Figure 8(b) exhibits the seismic intensity as assessed by AL-SM99. The AL-SM99 results effectively constrain the size of each intensity region. Along both sides of the rupture, the extent of each intensity region is nearly identical to that from ShakeMap. This method effectively mitigates the effects of aftershock anomalies that are not identified during the pre-processing phase. Owing to the influence

of aftershocks distributed at the far reaches of the fault, the fitted Lowess curve is slightly longer than the actual rupture, resulting in the assessed intensity range being slightly longer along the fault strike. However, the overall assessment outcome is acceptable. Although it is impossible to precisely depict the rupture of a complex fault system, we were still able to determine the extent of the affected region within a few hours of the earthquake based on the results of a good fit to the overall strike and length of the fault

rupture.

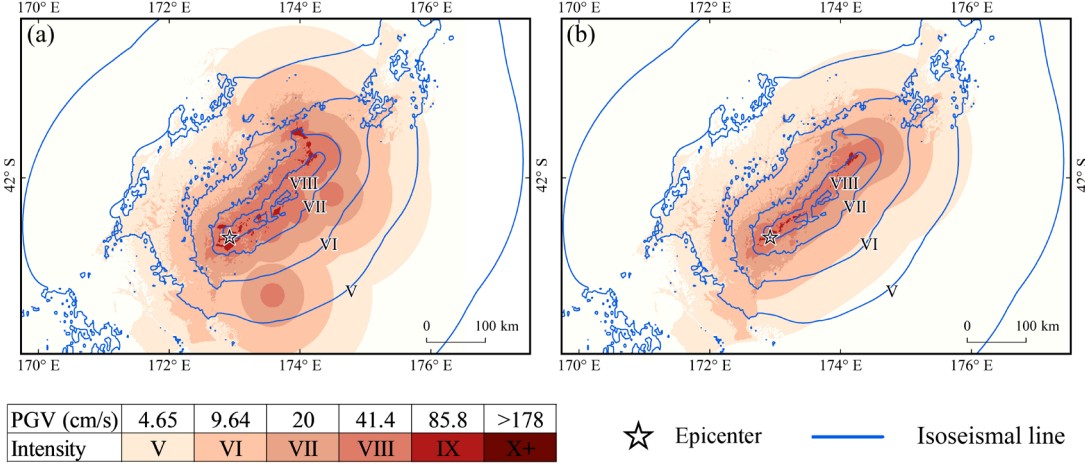

**Figure 8:** Intensity assessment of the 2016 Kaikōura Mw 7.8 earthquake using (a) pre-processed aftershocks and (b) Lowess results.

### 3.1.3 AL-SM99 latest application cases

According to the USGS report, Mw 7.8 and Mw 7.5 earthquakes struck on February 6, 2023, in the Kahramanmaraş region of Turkey. The death toll has reached 52,700 people in Turkey and Syria as of March 5, 2023, and caused an estimated US$89.2 billion in property damage in both countries, making it the deadliest natural disaster in modern Turkish history (Wikipedia, 2023). The causative fault for the

Mw 7.8 earthquake is located along the N60 striking East Anatolian Fault and continues towards the Dead Sea Fault and the N25 striking Karazu Fault, the causative fault for the Mw 7.5 earthquake is located north of the previous, along the N100 striking Sürgü-artak Fault (Provost et al., 2023). We conducted an intensity assessment immediately following the earthquake by collecting aftershock sequences from the Regional Earthquake-Tsunami Monitoring Center (http://www.koeri.boun.edu.tr/sismo/2/tr/) within 2 hours of both earthquakes, assessing the intensity of both earthquakes using AL-SM99 (Fig. 9). The intensity assessments of both events were consistent with the ShakeMap intensity maxima, at level IX. The results of the Lowess fit to the aftershock sequence revealed a bilateral rupture pattern for both earthquakes, providing a reference for examining the rupture of the source using physical inversion. In particular, the seismic intensity of the Mw 7.8 earthquake estimated using the Lowess results is highly consistent with that estimated using the inverse projection results (Chen et al., 2023). The extent of the affected region as determined by AL-SM99 is nearly identical to that of the recently updated intensity version of ShakeMap, accurately identifying a portion of the intensity anomalies. The shape of the Intensity VIII–IX region reflects the shape characteristics of the two causative faults; however, the intensity of the Mw 7.5 earthquake in the northwest is overestimated. Owing to the close proximity of the two earthquakes, we speculate that the fault and secondary faults in the region were activated by the superposition of the two earthquakes and produced more aftershocks, which were included in the intensity assessment of the second earthquake and were not judged as outliers, resulting in an overestimation of the intensity. In general, however, the intensity assessment results of the two earthquakes can provide reasonable early estimates of the extent of the disaster area, and the output of fine ground motion grid data can provide support for estimating casualties, property damage, etc. The application of AL-SM99 to these two earthquakes further demonstrates the applicability and dependability of the results.

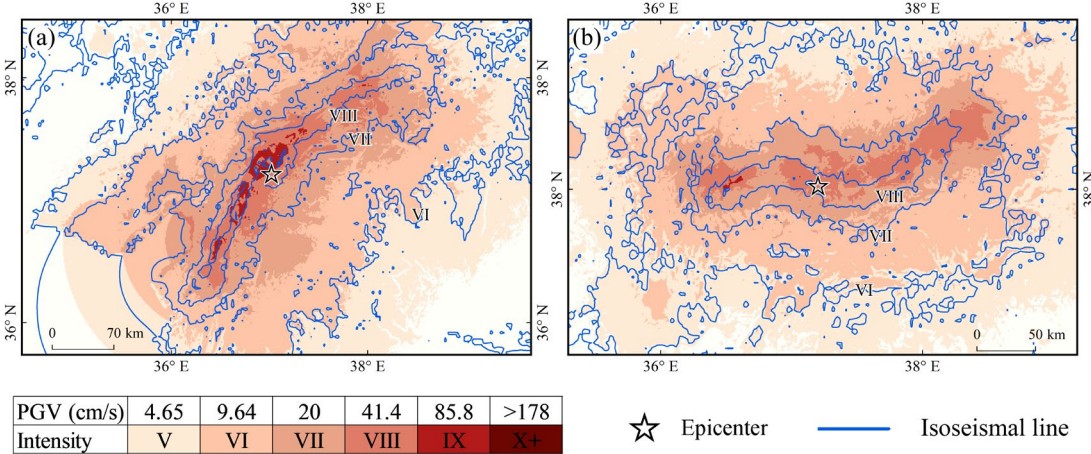

| PGV (cm/s) | 4.65 | 9.64 | 20 | 41.4 | 85.8 | >178 |
|------------|------|------|-----|------|------|------|
| Intensity  | V    | VI   | VII | VIII | IX   | X+   |

☆ Epicenter    —— Isoseismal line

**Figure 9:** Seismic intensity assessment results of the (a) 2023 Pazarcik (Kahramanmaraş) Mw 7.8 earthquake and (b) 2023 Elbistan (Kahramanmaraş) Mw 7.5 earthquakes. The isoseismal lines used for comparison are the outcomes of an evaluation of ShakeMap versions 15 and 9, respectively.

### 3.2 Lowess-fit results and fault rupture

Shortly after the mainshock, the aftershock sequence could outline the fundamental characteristics of the mainshock rupture surface (Kisslinger, 1996). We focused on the length and direction of the fault rupture in this study. Figure 10 exhibits the spatial distribution of the rupture trajectories and Lowess-fit curves. Table 3 compares the lengths and orientations of the fault rupture and fitted curves, and empirical rupture lengths are estimated using the Wells rupture formula (Wells and Coppersmith, 1994). The linear directional mean is a tool for measuring the direction or orientation of a set of lines; it determines the direction based solely on the starting and ending points of the lines (Mitchell, 2005). It was used in this study to determine the average orientation of the main rupture tracks.

In the Wenchuan, Baja California, Kaikōura, Ridgecrest, Maduo, Pazarcik and Elbistan earthquakes, the Lowess-fitted curves matched the actual fault rupture well in terms of linear directional mean and could accurately depict the orientation of the main rupture. Overall, the length of the Lowess-fitted curve is longer than the actual rupture. For certain earthquakes, the lengths of the fitted curves are closer to the lengths calculated by the empirical equations than to the actual ruptures. Because aftershocks are densely distributed at the tips of the fault (Hu et al., 2013), and the linear directional mean is only governed by the starting and ending points of the curve, the fitted curve has good similarity with the linear average direction of the actual rupture. The aftershocks that occurred at the tips of the fault exist at a certain

distance from the causative fault (Ozawa and Ando, 2021), causing the fitted curve to be longer than the surface rupture.

The quality of the aftershock catalogue has a major impact on the fitting results. The data source used to pick the aftershock events, the aftershock event location method, and the aftershock sequence selection method are all directly related to the spatial location of the aftershock sequence employed in AL-SM99, which further influences the length and direction of the fitted curve. (Liao et al., 2021; Zhao et al., 2022b). To ensure the accuracy of the fitted curves and enhance the temporal efficiency of assessing seismic

intensities, the number of aftershocks is important. Although we cannot currently provide an exact minimum standard for the number of aftershocks required for AL-SM99, it may be difficult to obtain significant results for less than 20 aftershocks, as in the case of the 2010 Palu earthquake. Moreover, the complexity of the fault system in the seismogenic region, such as its fractal dimension, has an effect on the outcomes (Nanjo and Nagahama, 2000; Baranov et al., 2022). When the fault strike near the

mainshock is known to be relatively clear and the fault system is simple, Lowess is a more appropriate method for analysing aftershock events.

**Table 3:** Comparison of Lowess-fit results for 11 earthquakes with actual surface rupture and results from the Wells' empirical model. The fitting length and linear directional mean were measured using ArcGIS.

| Date | Location | Magnitude | Rupture Length(km) | Fitting Length(km) | Wells result | Rupture direction | Fitting direction | Aftershocks |
|------|----------|-----------|--------------------|--------------------|--------------|-------------------|-------------------|-------------|
| 20051008 | Kashmir (Pakistan) | 7.6 | 70 | 138.77 | 135.25 | 306.50 | 335.62 | 54 |
| 20080512 | Wenchuan (China) | 7.9 | 275 | 340.88 | 238.26 | 39.97 | 40.60 | 43 |
| 20100404 | Baja California (Mexico) | 7.2 | 120 | 151.37 | 66.29 | 323.23 | 320.49 | 60 |
| 20111023 | Van (Turkey) | 7.1 | 31.95 | 60.39 | 52.64 | 11.12 | 3.86 | 46 |
| 20160415 | Kumamoto (Japan) | 7.0 | 34 | 164.75 | 43.94 | 35.58 | 40.65 | 538 |
| 20161113 | Kaikōura (New Zealand) | 7.8 | 184 | 242.88 | 197.28 | 31.35 | 30.52 | 106 |
| 20180928 | Palu (Indonesia) | 7.5 | 161.83 | 120.04 | 122.82 | 101.56 | 127.25 | 18 |
| 20190706 | Ridgecrest (US) | 7.1 | 63.42 | 62.11 | 53.97 | 137.60 | 137.94 | 105 |
| 20210521 | Maduo (China) | 7.3 | 99.5 | 158.42 | 81.42 | 346.43 | 347.93 | 70 |
| 20230206 | Pazarcik (Turkey) | 7.8 | 407.60 | 250.08 | 227.58 | 38.82 | 38.51 | 27 |
| 20230206 | Elbistan (Turkey) | 7.5 | 188.92 | 230.08 | 122.82 | 7.93 | 7.09 | 24 |


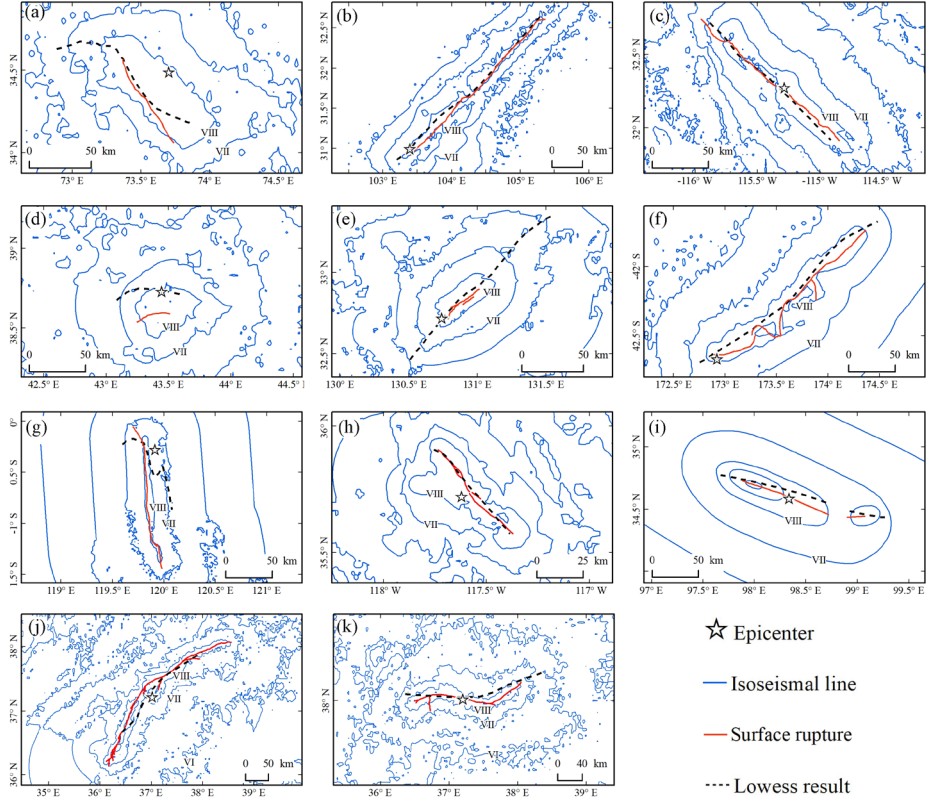

**Figure 10:** Comparison of Lowess-fitted curves with actual surface rupture for the (a) 2005 Kashmir Mw 7.6; (b) 2008 Wenchuan Mw 7.9; (c) 2010 Baja California Mw 7.2; (d) 2011 Van Mw 7.1; (e) 2016 Kumamoto Mw 7.0; (f) 2016 Kaikōura Mw 7.8; (g) 2018 Palu Mw 7.5; (h) 2019 Ridgecrest Mw 7.1; (i) 2021 Maduo Mw 7.3; (j) 2023 Pazarcik Mw 7.8; and (k) 2023 Elbistan Mw 7.5 earthquakes. Fault rupture traces were extracted from relevant literature or downloaded from ShakeMap, then digitised in ArcGIS (Kaneda et al., 2008; Li et al., 2008; Fletcher et al., 2014; Liu et al., 2015; Toda et al., 2016; Shi et al., 2019; Zhang et al., 2021; Reitman et al., 2023).

We gathered data on the estimated source rapture of the back-projection algorithm for the Wenchuan earthquake (Chen et al., 2022a). Using the same technique, a set of results reflecting the surface rupture of the 2016 Kaikura Mw 7.8 earthquake was calculated using waveform data from high sensitivity seismograph network in Japan. Both the Lowess and back-projection results show rupture directions similar to those indicated by the long axis of the isoseismal line in the area with intensity VIII of the Wenchuan earthquake, but the former estimates a longer rupture length (Fig.11(a)). Furthermore, the back-projection results reveal more details concerning the rupture. For example, the back-projection results indicate a possible fracture near the IX-degree intensity anomaly in the long-axis direction. This method has also demonstrated benefits in determining the intensity anomaly area for the 2022 Maduo Mw7.3 earthquake (Chen et al, 2022b). As a nonparametric method, the points fitted by Lowess are clearly distributed along a curve. However, when the fault system in the seismogenic region is complex,

the dominant orientation of the rupture traced using the back-projection method may be problematic

(Fig.11(b)). A clear guide to array data selection may be required when using the back-projection method, and we recognize that the results of array data calculations are more accurate when the appropriate region is chosen (Wang and Hutko, 2018). Aftershocks that have been relocated can be used to determine rupture fault trajectories, and their combination with inverse projection techniques has been applied to determine transient shear ruptures (Li et al., 2019; Cheng et al., 2023). These two methods could be cross-referenced

in application for more accurate intensity evaluation results overall.

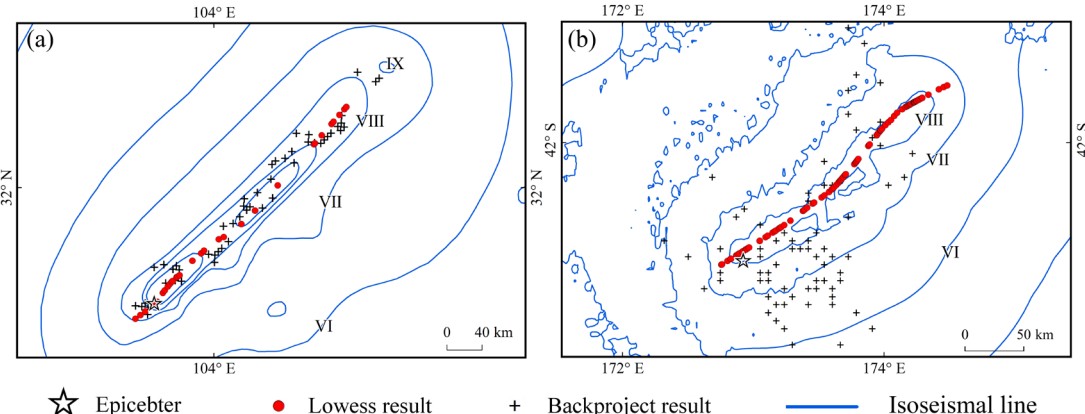

**Figure 11**:Comparison of surface rupture results obtained using the Lowess and back-projection methods for the (a) 2008 Wenchuan Mw 7.9 and (b) 2016 Kaikōura Mw 7.8 earthquakes.

## 4 Discussion

**4.1 Stability of AL-SM99**

In essence, we have explored the possibility of exploiting early aftershock data through Lowess and found that this method achieved fairly accurate earthquake intensity assessment results. The quality of aftershock data (number of aftershocks, accuracy of aftershock localisation, and magnitude of aftershocks) controls the robustness of the method to a certain extent. The aftershock data observed by

conventional stations are sufficient for our method (Zhao et al., 2022a, 2022b); however, good data quality could produce more accurate assessment results. In addition, we used a robust version of Lowess, whose parameter $f$ controls the degree of smoothing and is the only one that needs to be adjusted based on the data properties during the implementation of this algorithm. The criterion for determining $f$ is to choose a value that is as large as possible to minimize the variability in the smoothed points without

damaging the data pattern. The larger the setting of parameter $f$, the smoother the fitting result is; its

recommended value ranges from 0.2 to 0.8. Without knowing the data characteristics, it is reasonable to take 0.5 as the starting value (Cleveland, 1979). The value of $f$ determines the size of the local range over which the weighted linear regression is performed. When $f$ is reduced to a small value, fewer aftershocks are used to determine the fitted values in that range, causing the fitted curve to shift toward aftershocks that are more spatially dispersed. The fitted curve deviates significantly from the position of the fault rupture track. Therefore, the accuracy of our intensity assessment is more controlled by smoothness $f$.

Taking the 2008 Wenchuan Mw 7.9 and 2016 Kaikōura Mw 7.8 earthquakes as examples, when $f$ is set to a lower value, the fitting results are more affected by the local data, particularly the aftershocks that were spatially distributed at a significant distance could interfere with the calculation of GMPE and affect the accuracy of earthquake intensity assessment (Fig. 12). When $f$ exceeds a certain value, the fitted curve becomes smoother more concentrated in a specific region. The influence of aftershocks with abnormal spatial location on the fitting effect gradually decreases; this stable result reflects the spatial characteristics of the mainshock rupture and can be used in the calculation of SM99. For $f = 0.5$, the fitted curve is in the stable region. The number and spatial distribution of aftershocks have a significant impact on the value of parameter $f$. In a real-world seismic emergency, the parameters can be adjusted based on the characteristics of the acquired data as well as expert experience to produce a curve that is more consistent with the rupture characteristics.

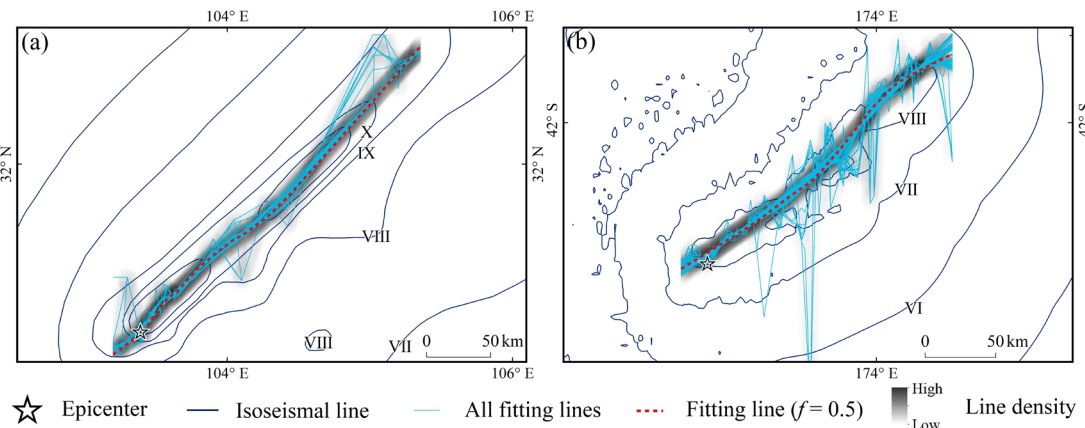

**Figure 12:** Lowess-fitted curves and line densities for different values of smoothness f for the (a) 2008 Wenchuan Mw 7.9 and (b) 2016 Kaikōura Mw 7.8 earthquakes. A total of 99 curves were fitted using Lowess in steps of 0.01 from 0.01 to 0.99.

**4.2 Conditions for the application of AL-SM99**

Previous studies have utilised known fault geological data and far-field array back-projection results for

SM99 GMPE calculations (Zhang et al., 2021; Chen et al., 2022). Our research increases the number of

415 data sources used for such calculations. As AL-SM99 requires a certain number of aftershocks for

accuracy, the earthquake to which the method is applied must be accompanied by a certain size aftershock

or be a swarm-type earthquake.

**Table 4:** Average residuals of $PGV_{vs30}$ predictions within 100 km for 23 earthquakes, this PGV results obtained using AL-SM99. The dominant slip types were identified based on the moment tensor results obtained from the

420 ShakeMap website.

| Date | Country | Magnitude | Residual (log₁₀(obs./calc.)) | Slip type | Space distribution |
|---|---|---|---|---|---|
| 20130816 | New Zealand | 6.5 | 0.3737 | Strike-slip fault | random |
| 20110411 | Japan | 6.6 | 0.2447 | normal fault | cluster |
| 20161030 | Italy | 6.6 | 0.0341 | normal fault | cluster |
| 20180905 | Japan | 6.6 | 0.2531 | reverse thrust | cluster |
| 20220107 | China | 6.6 | 0.1999 | Strike-slip fault | cluster |
| 20220905 | China | 6.6 | 0.1748 | Strike-slip fault | cluster |
| 20001006 | Japan | 6.7 | 0.1679 | Strike-slip fault | cluster |
| 20080613 | Japan | 6.9 | 0.0307 | reverse thrust | cluster |
| 20180504 | America | 6.9 | -0.1511 | reverse thrust | random |
| 20030526 | Japan | 7.0 | 0.0786 | reverse thrust | cluster |
| 20100903 | New Zealand | 7.0 | -0.1179 | Strike-slip fault | random |
| 20160415 | Japan | 7.0 | -0.1025 | Strike-slip fault | cluster |
| 20201030 | Greece | 7.0 | 0.3235 | normal fault | cluster |
| 20181130 | America | 7.1 | 0.1723 | normal fault | cluster |
| 20190706 | America | 7.1 | -0.0472 | Strike-slip fault | cluster |
| 20171112 | Iraq | 7.4 | 0.1913 | reverse thrust | dispersed |
| 20230206 | Turkey | 7.5 | 0.2654 | Strike-slip fault | cluster |
| 20051008 | Pakistan | 7.6 | 0.1355 | reverse thrust | random |
| 20150425 | Nepal | 7.8 | -0.0476 | reverse thrust | cluster |
| 20161113 | New Zealand | 7.8 | -0.0392 | reverse thrust | cluster |
| 20230206 | Turkey | 7.8 | 0.1557 | Strike-slip fault | random |
| 20080512 | China | 7.9 | -0.1693 | reverse thrust | random |
| 20150916 | Chile | 8.3 | 0.0827 | reverse thrust | cluster |

According to the empirical relationship, large earthquakes are frequently accompanied by significant

surface rupture (Bonilla et al., 1984; Wells and Coppersmith, 1994). In this study, the $PGV_{vs30}$ calculated

using the Lowess-fit results and VS30 data for earthquakes with Mw ≥ 7.0 was superior to that calculated

for earthquakes with Mw 6.5–6.9 (Table 4). The b-value describes the number of small earthquakes that occur following every large earthquake. A significant increase in the b-value was also discovered for earthquakes with Mw ≥ 7.0 in a study of the effect of mainshocks on the aftershock size distribution (Gulia et al., 2018; Van Der Elst, 2021). There was no significant correlation between this method and the fault slip types or the degree of spatial aggregation of aftershocks, and the type of source mechanism appeared to have no significant effect on the results (Kagan, 2002). When the magnitude is small, we can directly use SM99 to calculate PGV and PGA after excluding outliers, or we can add a buffer a certain distance from the fitted curve to select the aftershock data for calculation (Ozawa and Ando, 2021; Zhao et al., 2023). If necessary, aftershocks that are not deemed outliers can be manually removed with expert experience. GMPEs also influence the calculation of small-magnitude earthquakes to a certain degree (Chen et al., 2022a).

Aftershocks accumulate after an earthquake, and the probability of an aftershock occurs in a hyperbolic relationship with time (Omori, 1894; Utsu, 1961; Guglielmi and Zavyalov, 2018). As previously mentioned, the number of aftershocks has a significant impact on the Lowess fitting results. In this study, the first quartile, median, and third quartile of aftershock number were 43, 70, and 116.5, respectively. Conservatively, using more than 40 aftershocks for the assessment is likely to yield more reliable results. Even foreshocks may be the result of the earthquake nucleation process (Dodge et al., 1996). Therefore, foreshocks can be considered for inclusion in the raw data to increase the amount available.

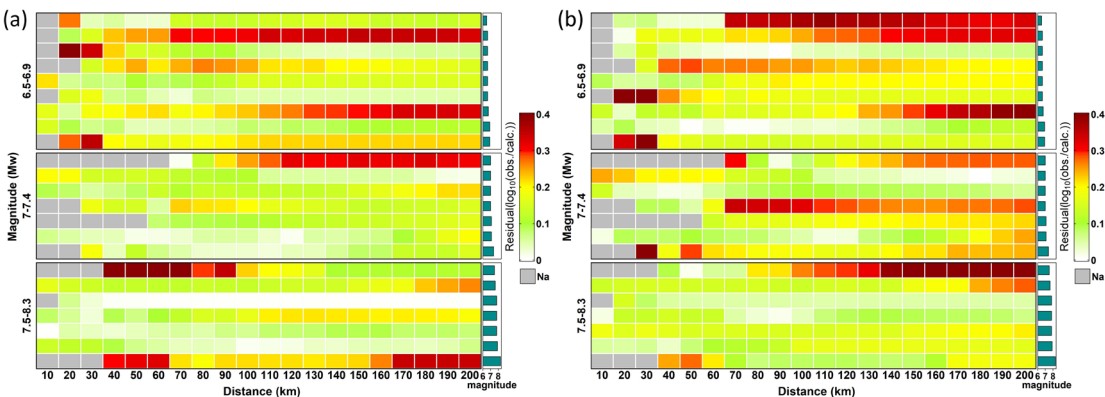

**Figure 13:** Heat map of the average residuals of predicted (a) PGA and (b) PGVvs30 values for 23 earthquakes, which have good station records. A residual value is calculated for every 10 km increase in the range of 200 km from the epicentre, and the corresponding colour is assigned to the corresponding position in the graph. The magnitudes were divided into three groups. Each row represents an earthquake, and the histogram on the left displays the associated magnitude.

The model prediction results are credible if the aftershocks accurately reflect the information of the causative faults as the input data of SM99 GMPE. The average residuals of the PGVvs$_{30}$ and PGA predicted values for the 23 earthquakes were between -0.4 and 0.4 (Fig. 13). With increasing magnitude, the residuals of ground motion prediction decrease significantly. The residuals of ground-motion predictions for earthquakes with magnitudes of 7.5–8.3 are superior to those of the other two subgroups, whereas the residuals of Mw 6.0–6.5 are higher. This implies that the method is more applicable in large-magnitude earthquakes. For many earthquakes shown in Fig. 13, the residuals of the ground motion prediction results increase with distance, indicating the advantage in determining the extent of the hardest hit areas.

ShakeMap has established itself as the authoritative global platform for distributing and sharing earthquake information and has played a critical role in documenting many major earthquakes and geological disasters (Worden et al., 2020). However, in certain regions, particularly those with limited access to station monitoring data, the seismic intensity estimated by ShakeMap may differ significantly from the actual situation. For example, the 2021 Maduo Mw 7.3 earthquake shown in Figure 14 produced a northwest-southeast oriented surface rupture and an intensity anomaly zone southeast of the epicentre (Zhang et al., 2021; Chen et al., 2022b). The estimated direction of the ShakeMap high intensity regions in Fig. 14(b) was close to the west-east distribution, and the range of intensity VIII was significantly different from the actual intensity. Aftershock fitting results recorded within 2 hours of the mainshock can be used to determine the causative fault and provide a reference for examining the results of finite fault and back projection. GMPEs possess a strong regional identity. The selection of appropriate GMPEs within a seismogenic region allows for a more precise estimation of earthquake intensity.

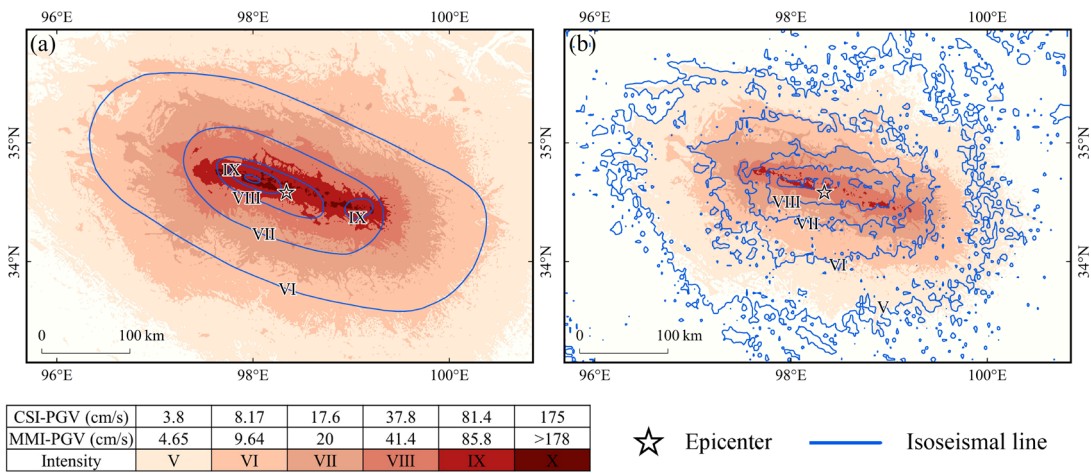

**Figure 14:** Intensity assessment of the 2021 Maduo Mw 7.3 earthquake. Comparison of the seismic intensities determined by Lowess with those determined by (a) CEA and (b) ShakeMap.

**4.3 Time efficiency**

The primary goal in developing this method was to provide information services to response workers during the black box period of an earthquake emergency. We learned the following from the calculations of all the cases in this study and from actual earthquakes emergency work (Zhao et al., 2022b, 2023). If seismic stations in the seismogenic region are as sparse and uneven as those in western China, once the earthquake is determined to be suitable for use with AL-SM99, a reliable seismic intensity assessment map can be produced within 1–1.5 hours of the mainshock. The majority of the time required to produce the results is spent acquiring aftershock data, while processing the aftershock data requires only a few seconds, and the calculation and output of the maps require approximately five minutes. Taking the 2016 Kaikōura Mw 7.8 earthquake as an example, the fitted curve gradually lengthened with an increase in time, and the direction indicated by the beginning and end of the curve in different periods did not change significantly, although the curve changed locally in a more obvious manner (Fig. 15). This indicates that the curve fitted by Lowess is constantly and dynamically corrected as the number of aftershocks increases and that the use of aftershock sequences approximately 1.5 hours after the earthquake can sufficiently assess the seismic intensity distribution. This is consistent with the experience mentioned above.

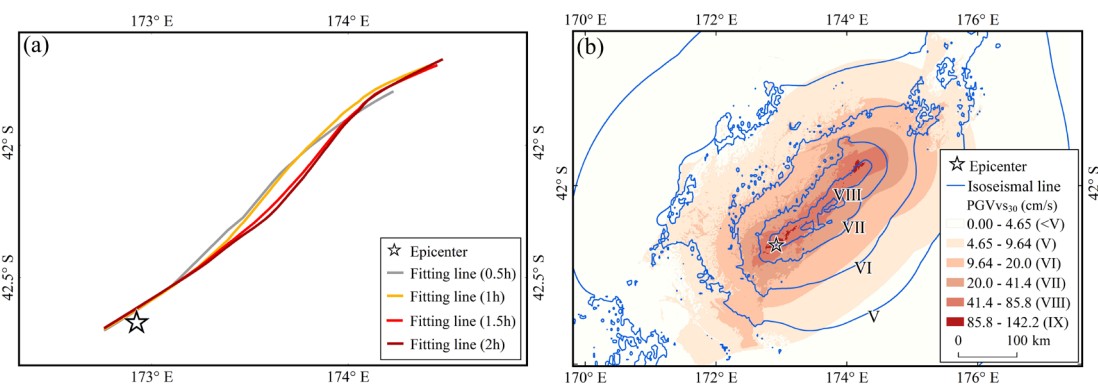

**Figure 15:** Lowess split-time fitting results of the 2016 Kaikōura Mw 7.8 earthquake. (a) Lowess fitting curves plotted at 0.5 h intervals; and (b) assessment of seismic intensity using aftershocks within 1.5 hours of the earthquake.

In areas with dense monitoring stations or those using artificial intelligence aftershock pickup methods, a large number of aftershock data can often be obtained in a short time. For example, in Japan, a large number of good aftershock records can significantly shorten the intensity assessment time (Fig. 16). Good data quality can shorten the intensity assessment results to within 30 min and greatly increase the

efficiency of the intensity assessment. Chen et al. (2022a) proposed a rapid assessment method that can generate intensity assessment maps within 30 min. The spatial location of the rupture trajectories obtained from the inversion of rupture processes for small magnitude earthquakes may be less satisfactory (Honda et al., 2011; Yao et al., 2019); however, for these earthquakes, the seismic intensities assessed using aftershock data may be more accurate (Kang et al, 2023). The AL-SM99-fitted curves of

the spatial distribution of aftershocks can be used as a cross-reference for the correction of the above inversion results, increasing the speed of the operation using both methods. For global earthquakes, when the magnitude reaches the trigger threshold of the ShakeMap system, the first version of the assessment results is generated through the original solution built into the system within minutes of the mainshock and is continuously updated as data are aggregated and accumulated (Worden et al, 2020). Thus, we have

considered combining AL-SM99 with aftershock monitoring to dynamically present intensity assessment results, because for earthquakes with small rupture scales, relying on the epicentre coordinate or a small number of aftershocks can provide useful shaking distribution estimates.

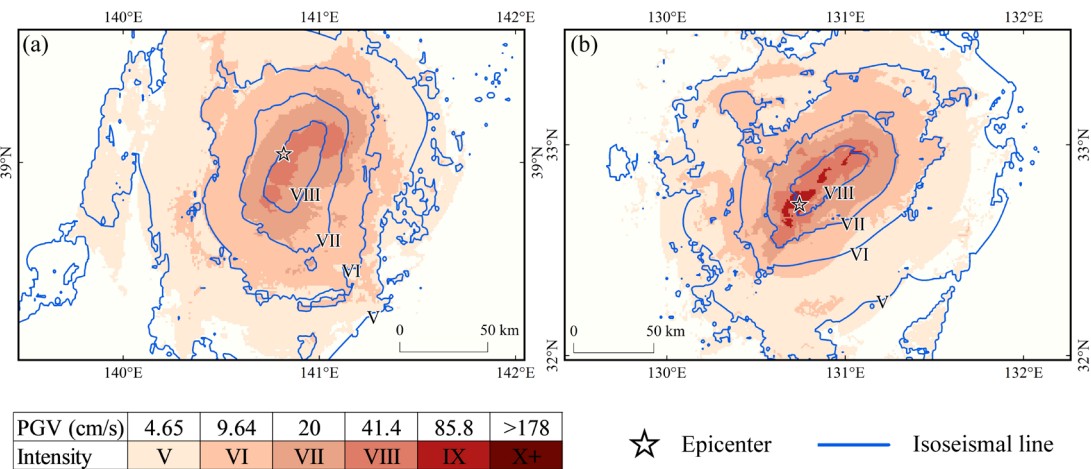

**Figure 16:** Results of a rapid assessment of seismic intensities. The seismic intensities of the (a) 2008 Iwate-Miyagi
Nairiku Mw 6.9 and (b) 2016 Kumamoto Mw 7 earthquakes were evaluated using aftershocks that occurred within 10 min. The latter manually removed distant aftershocks that were not identified as outliers.

## 5 Conclusions

In this study, we developed a method for evaluating seismic intensities based on aftershock data gathered within 2 hours of the mainshock. Aftershock sequences are treated as scatterplots, with Lowess fitting

applied to their longitude and latitude coordinate values. The result of the fit is used to roughly describe

the fault rupture trend, and the SM99 GMPE was used to calculate ground motion data. The PGV values were then converted to seismic intensity. The main conclusions are as follows:

1. The length and direction of the surface rupture can be roughly outlined by the early aftershock sequence following the mainshock. The fitted curves from Lowess are helpful for pinpointing the location of causative faults and rupture scales. When the fault system in the seismic region is clear and simple, the Lowess fitted curves can be used to accurately determine the location and length of the fault rupture. When the fault system is complex, Lowess results can still indicate the overall rupture trend and make reliable rupture scale judgments.

2. Lowess is suited for aftershock sequences of large magnitude earthquakes (Mw ≥7.0). The fitted curves are always slightly longer than the actual surface rupture, indicating that aftershocks occurred at a certain distance from the tips of the fault shortly after the mainshock (Ozawa and Ando, 2021). This method broadens the scope of application for early post-earthquake aftershock data.

3. Aftershocks frequently cause secondary damage to buildings in the affected region, resulting in greater economic losses or fatalities. The seismic intensity map based on the spatial distribution trend assessment of aftershock sequences could reflect the extent of the hardest-hit areas and regions where cause property damage and fatalities may occur.

4. When the listed conditions are met, the seismic intensities assessed using AL-SM99 can serve as a useful reference for early earthquake emergency response efforts. The outcomes of intensity assessment may also provide a basis for different perspectives in studying the radiative energy of earthquakes and locating causative faults. Obviously, selecting the appropriate GMPEs can produce more accurate intensity assessment results.

Notably, only the coordinate positions of the aftershocks are used when fitting the aftershock sequence with Lowess. A discrepancy remains between the fitted curve length, local trend, and the actual surface rupture. In future research, the type of the causative faults and geological context of the seismogenic regions will be considered, and empirical formulas such as Wells' surface rupture formula will be used for correction. It is beneficial to study the aftershock sequence relocation methods and the relationship between the spatial distribution of early aftershock sequences and causative faults. The application of Lowess to smoothing the spatial distribution trends of aftershock sequences over extended time periods is also of interest. AL-SM99 can dynamically generate intensity assessment results in conjunction with aftershock monitoring networks. Although the viability of aftershock prediction remains debatable, it is

possible to combine aftershock predictions and achieve rapid seismic intensity prediction (DeVries, et al., 2018; Mignan and Broccardo, 2019).

**Data availability.** The selection of earthquake cases, station lists, and seismic intensity vector files for this study were downloaded from https://earthquake.usgs.gov/data/shakemap/ (last access: 15 March 2023); aftershock data for earthquakes outside of China were downloaded from http://www.isc.ac.uk/iscbulletin/search/catalogue/ (last access: 16 August 2022); aftershock data for earthquakes that occurred in China were downloaded from http://earthquake.ckcest.cn/dzcestsc/ earthquake_tyml.html (last access: 16 August 2022 ).

**Financial support.** This research was supported by the Major Science and Technology Projects of Gansu Province (21ZD4FA011), and the National Key Research and Development Program of China (No. 2017YFB0504104).

**Author contributions.** WC conceptualised the project, acquired funding, and supervised the project. HZ performed the investigation, deployed the software and code, and edited the manuscript. WC and HZ developed the methodology and revised the manuscript. CZ and DK provided input and assistance in the improvement of the methodology.

**Competing interests.** The authors declare that they have no competing interest.

**Acknowledgements.** The procedures for screening and deleting outliers, Lowess smoothing, and calculation of residuals and other indicators were carried out in R. The seismic intensity maps and the effect maps of the fitted curves were carried out in ArcGIS software.

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
