# Peer review of "Rapid estimation of seismic intensities by analysing early aftershock sequences using the robust locally weighted regression program (Lowess)"

_Natural Hazards and Earth System Sciences, 2022_

## Referee Comment (RC2)

[referee-annotated manuscript omitted]

---

## Author Comment (AC1)

We would like to thank the anonymous referee for the thoughtful review of our manuscript and for giving these constructive comments and suggestions, which substantially helped us improve the quality of the paper. All the points that were raised have been adopted in the revised manuscript. We believe the new version of the manuscript has been significantly improved. Below is a point-by-point answer to the comments and suggestions raised by the reviewer.

**Major comments:**

1. Aftershock selection and illustration.
Number and locations of early aftershocks are critical in this algorithm. I am not sure how the early (within 2 h) aftershocks could accurately reveal the rupture pattern of earthquakes. I do recommend the authors make plots of the early aftershocks for the events shown in this manuscript. By doing this, readers can easily judge how the early aftershocks reflected the source dimension. Statistical analysis might be required to demonstrate this question.

**Reply:** In the new version of the manuscript, we have added temporal and spatial distribution plots of the early aftershock sequences of the Wenchuan Mw7.9 and Kaikōura Mw7.8 earthquakes in Section 2.2.2, as well as interpreted the insets. The added content is as follows: "The early aftershocks of these two earthquakes were mainly distributed along the direction of the surface rupture zone and within a certain range on both sides of the surface rupture zone, based on the spatial and temporal distribution of the aftershock sequences (Fig. 3). The Wenchuan earthquake's early aftershocks mostly occurred within 300 kilometers of the epicenter and were concentrated along the main rupture direction. And the early aftershocks of the Kaikōura earthquake were distributed within 200 kilometers of the epicenter and were relatively dispersed along the main rupture strike, which was primarily caused by the earthquake's complex fault system (Wallace et al., 2018). Early aftershocks in both cases exhibit a pattern of rupture in a single direction. Based on this, we believe that the fitting results in figure 2 can roughly depict the length and direction information of the earthquake rupture."

[Figure]

**Figure 3:** Spatial and temporal distribution of early aftershock sequences after the mainshock. (a) and (c) depict the spatial distribution of aftershocks of the 2008 Wenchuan Mw 7.9 earthquake and the 2016 Kaikōura Mw 7.8 earthquake, respectively. (b) and (d) depict the temporal distribution of aftershocks of the Wenchuan and Kaikōura earthquakes, respectively. The s in Log(s) denotes the time in seconds between the aftershock and the mainshock.

2. The authors have proven that the accuracy of the estimated intensity map by this method by comparing it with other results. That is good. We can estimate the damage levels in space. But the time efficiency is less discussed or demonstrated. How fast you could deliver this result? And comparing it with other approaches would greatly enhance the importance of this work.

**Reply:** Some discussion of time efficiency and comparisons with other methods are added to Section 4.2, and the corresponding references were added. The added contents are as follows: "The primary goal of developing this method is to provide information services to response workers during the black box period of an earthquake emergency. Lessons learned from the calculations of all the cases in this study and from actual earthquakes emergency work (Zhao et al., 2022b; Zhao et al., 2023). If seismic stations in the seismogenic region are as sparse and uneven as those in western China, once the earthquake is determined to be suitable for use with AL-SM99, a reliable seismic intensity assessment map can be produced within 1-1.5 h of the mainshock. The majority of the time required to produce the results is spent acquiring aftershock data, while processing the aftershock data takes only a few seconds and the calculation and output of the maps takes about five minutes."

"Chen et al (2022a) proposed a rapid assessment method that can generate intensity assessment maps within 30 minutes. The spatial location of the rupture trajectories obtained from the inversion of rupture processes for earthquakes of small magnitudes may be less satisfactory (Honda et al., 2011; Yao et al., 2019). In small-magnitude earthquakes, however, seismic intensities assessed using aftershock data may be more accurate (Kang et al, 2023). The AL-SM99-fitted curves of the spatial distribution of aftershocks can be used as a cross-reference for the correction of the above inversion results, speeding up the operation using both methods. For global earthquakes, when the magnitude reaches the trigger threshold of the ShakeMap system, it will generate the first version of the assessment results through the original solution built into the system within minutes after the earthquake, and will be continuously updated as data is aggregated and accumulated (Worden et al, 2020). It inspires us to combine AL-SM99 with aftershock monitoring to dynamically present intensity assessment results, since for earthquakes with small rupture scales, relying on the epicenter coordinate or a small number of aftershocks can provide very useful shaking distribution estimates."

Reference:

"Honda, R., Yukutake, Y., Ito, H., Harada, M., Aketagawa, T., Yoshida, A., Sakai, S. I., Nakagawa, S., Hirata, N., and Obara, K.: A complex rupture image of the 2011 off the pacific coast of Tohoku earthquake revealed by the meso-net, Earth Planet Sp., 63(7), 583–588, https://doi.org/10.5047/eps.2011.05.034, 2011."

"Yao, Q., Wang, D., Fang, L. H., and Mori, J.: Rapid estimation of magnitudes of large damaging earthquakes in and around Japan using dense seismic stations in China, Bull. Seismol. Soc. Am. 109, 2545–2555. https://doi.org/10.1785/0120190107, 2019."

3. To better validate the accuracy of the source dimension estimated from the early aftershocks, the authors could compare your results with source ruptures, at least for large earthquakes. I believe there are many cases that can be utilized for such comparison.

**Reply:** Nine earthquakes with Mw≥7.0 were used as an example in Section 3.2.Our results were compared to surface rupture lengths calculated using an empirical formula for wells and those documented in the literature, and the average linear direction of surface rupture was calculated using ArcGIS software (Table 2). Subject to the conditions of use, the results of our method's fitting can provide reasonably accurate information on the length and direction of surface rupture. In addition, we include a comparison with the back-projection results of Chen et al (2022a) in this section, which supports our conclusions. However, since Lowess is essentially a nonparametric regression method that ignores the complex physical relationships contained in the aftershock sequence, we believe its results cannot fully replace those obtained through physical means (e.g., back-projection techniques). But the different methods can be cross-referenced to make further corrections to the results. The added content is as follows:

"We gathered the source rupture data from Chen et al (2022a) using the back-projection technique. And using the same technique, a set of results reflecting the surface rupture of the 2016 Kaikura Mw7.8 earthquake was calculated, using waveform data from high sensitivity seismograph network in Japan. Both the Lowess and back-projection results show rupture directions similar to those indicated by the long axis of the isoseismic line in the area with intensity VIII of the Wenchuan earthquake, but the former estimates a longer rupture length (Fig.11(a)). Furthermore, the back-projection results reveal more details about the rupture. For example, the back-projection results point to a possible fracture near the IX-degree intensity anomaly in the long-axis direction. This method has also demonstrated benefits in determining the intensity anomaly area in the application of the 2022 Maduo Mw7.3 earthquake (Chen et al, 2022b). As a nonparametric method, the points fitted by Lowess are clearly distributed along a curve. However, when the fault system in the seismogenic region is complex, the dominant orientation of the rupture traced using the back-projection method may be problematic(fig.11(b)). A clear guide to array data selection may be required when using the back-projection method, and we recognize that the results of array data calculations will be more accurate if the appropriate region is chosen. We believe that the two methods can be cross-referenced in their application to obtain more accurate intensity assessment results.

[Figure]

**Figure 11:** Comparison of surface rupture results obtained using the lowess and inverse projection methods for the

(a) 2008 Wenchuan Mw 7.9 and (b) 2016 Kaikōura Mw 7.8 earthquake."

4. Comparison of your results with Chen et al. (2022a, b) that you already cited in this work is also beneficial.

**Reply:** We have added a comparison with Chen et al's (2022a,b) work to both the examination of the source rupture results and the discussion of time efficiency, which adds to the richness of our manuscript.The additions are mentioned above in the responses to major comment 2 and 3.

**Minor comments:**

1. The English needs improvements.
Line 10, mainshocks
Line 13, of 59 M XXX~XXX earthaukes that occurred from 2000-2022
Line 21, Our study suggest that with early accessible aftershocks, we are able to rapidly determine the rupture fault plane (s), thus have better estimae of the seismic intensities.
Line 44, of an earthquake is limited,
Line 47, after earthquakes
Line 94, We selected Mw ≥ 6.6 shallow earthquakes that occurred during 2000-2022 in this study.
...

**Reply:** We have checked the language errors in the manuscript and polished it.

---

## Author Comment (AC2)

**1. The new illustrations**

[Figure]

**Figure 3:** Spatial and temporal distribution of early aftershock sequences after the mainshock. (a) and (c) depict the spatial distribution of aftershocks of 2008 Wenchuan Mw7.9 earthquake and 2016 Kaikōura Mw 7.8 earthquake, respectively. (b) and (d) depict the temporal distribution of aftershocks of Wenchuan and Kaikōura earthquake, respectively. The s in Log(s) denotes the time in seconds between the aftershock and the mainshock.

[Figure]

**Figure 11:** Comparison of surface rupture results obtained using the lowess and inverse projection methods for the (a) 2008 Wenchuan Mw 7.9 and (b) 2016 Kaikōura Mw 7.8 earthquake.

**2. Examine our method using examples**

**(1) 2022 Luding Mw 6.6 earthquake**

[Figure]

(a) Seismic intensities evaluated using AL-SM99. (b) Evaluation of seismic intensities using the modified AL-SM99 strategy. The solid blue line represents the China Earthquake Administration's intensity survey results.

(2) 2023 Turkey Mw 7.8 earthquake

(a) Seismic intensities evaluated using AL-SM99. Data on aftershocks gathered within 2 hours of

the earthquake. Aftershock data within two hours downloaded from REGIONAL EARTHQUAKE-TSUNAMI MONITORING CENTER (RETMC) in Turkey (http://www.koeri.boun.edu.tr/sismo/2/tr/). (b) Seismic intensity result released by USGS (Version 12) (https://earthquake.usgs.gov/earthquakes/eventpage/us6000jllz/shakemap/intensity). (c) Comparison of AL-SM99-evaluated seismic intensity result with USGS intensity result. The solid blue line represents intensity results from the USGS.

(3) 2023 Turkey Mw 7.5 earthquake

**土耳其Mw7.5级地震动强度（PGV）评估图V1.0**

[Figure]

[Figure]

[Figure]

(a) Seismic intensities evaluated using AL-SM99. Data on aftershocks gathered within 2 hours of the earthquake. Aftershock data within two hours downloaded from REGIONAL EARTHQUAKE-TSUNAMI MONITORING CENTER (RETMC) in Turkey (http://www.koeri.boun.edu.tr/sismo/2/tr/). (b) Seismic intensity result released by USGS (Version 8). (https://earthquake.usgs.gov/earthquakes/eventpage/us6000jlqa/shakemap/intensity) (c) Comparison of AL-SM99-evaluated seismic intensity result with USGS intensity result. The solid blue line represents intensity results from the USGS.

---

## Author Response (AR1)

We would like to thank the anonymous referees for the thoughtful review of our manuscript, as well as their insightful comments and suggestions, which helped substantially improve the quality of the paper. We are extremely grateful to the editors for their serious and responsible attitude towards the manuscript, which has been helpful to the timely revision of the manuscript.

Almost all the points that were raised have been adopted in the revised manuscript. We believe the new version has been significantly improved. We have revised the paper considering all the comments, which are discussed below point-by-point. In addition, minor errors have been corrected in the text. In the marked-up version of the manuscript, revisions are highlighted in red.

**Major comments (RC1):**

1. Aftershock selection and illustration.

Number and locations of early aftershocks are critical in this algorithm. I am not sure how the early (within 2 h) aftershocks could accurately reveal the rupture pattern of earthquakes. I do recommend the authors make plots of the early aftershocks for the events shown in this manuscript. By doing this, readers can easily judge how the early aftershocks reflected the source dimension. Statistical analysis might be required to demonstrate this question.

**Reply:** In the new version of the manuscript, we have added spatiotemporal distribution plots of the early aftershock sequences of the Wenchuan Mw 7.9 and Kaikōura Mw 7.8 earthquakes in Section 2.2.2, as well as interpreted the insets (lines 163–177). The added content is as follows:

"The early aftershocks of these two events were distributed along the surface rupture zone and within a certain range on both sides of the zone, based on the spatiotemporal distribution of the aftershock sequences (Fig. 3). The early aftershocks of the Wenchuan earthquake occurred within 300 km of the epicentre and were concentrated along the main rupture direction, and those of the Kaikōura earthquake were distributed within 200 km of the epicentre and relatively dispersed along the main rupture strike, which was primarily caused by the complex fault system (Wallace et al., 2018). Early aftershocks in both cases ruptured in a single direction. Based on this, we believe that the fitting results in Fig. 2 can depict the length and direction of the earthquake rupture."

[Figure]

**Figure 3:** Spatiotemporal distribution of early aftershock sequences after the mainshock. (a) and (c) Spatial distribution of aftershocks of 2008 Wenchuan Mw7.9 and 2016 Kaikōura Mw 7.8 earthquakes. (b) and (d) Temporal distribution of aftershocks from Wenchuan and Kaikōura earthquakes. The s in Log(s) denotes the time in seconds between the aftershock and mainshock.

2. The authors have proven that the accuracy of the estimated intensity map by this method by comparing it with other results. That is good. We can estimate the damage levels in space. But the time efficiency is less discussed or demonstrated. How fast you could deliver this result? And comparing it with other approaches would greatly enhance the importance of this work.

**Reply:** A discussion of time efficiency and comparisons with other methods have been added to Section 4.3, and the corresponding references were added (lines 479-486; lines 500-512). The added contents are as follows:

"The primary goal in developing this method was to provide information services to response workers during the black box period of an earthquake emergency. We learned the following from the calculations of all the cases in this study and from actual earthquakes emergency work (Zhao et al., 2022b, 2023). If seismic stations in the seismogenic region are as sparse and uneven as those in western China, once the earthquake is determined to be suitable for use with AL-SM99, a reliable seismic intensity assessment map can be produced within 1–1.5 hours of the mainshock. The majority of the time required to produce the results is spent acquiring aftershock data, while processing the aftershock data requires only a few seconds, and the calculation and output of the maps require approximately five minutes."

"Chen et al. (2022a) proposed a rapid assessment method that can generate intensity assessment maps within 30 min. The spatial location of the rupture trajectories obtained from the inversion of rupture processes for small magnitude earthquakes may be less satisfactory (Honda et al., 2011; Yao et al., 2019); however, for these earthquakes, the seismic intensities assessed using aftershock data may be more accurate (Kang et al, 2023). The AL-SM99-fitted

curves of the spatial distribution of aftershocks can be used as a cross-reference for the correction of the above inversion results, increasing the speed of the operation using both methods. For global earthquakes, when the magnitude reaches the trigger threshold of the ShakeMap system, the first version of the assessment results is generated through the original solution built into the system within minutes of the mainshock and is continuously updated as data are aggregated and accumulated (Worden et al, 2020). Thus, we have considered combining AL-SM99 with aftershock monitoring to dynamically present intensity assessment results, because for earthquakes with small rupture scales, relying on the epicentre coordinate or a small number of aftershocks can provide useful shaking distribution estimates."

References added:

Honda, R., Yukutake, Y., Ito, H., Harada, M., Aketagawa, T., Yoshida, A., Sakai, S. I., Nakagawa, S., Hirata, N., and Obara, K.: A complex rupture image of the 2011 off the pacific coast of Tohoku earthquake revealed by the meso-net, Earth Planet Sp., 63(7), 583–588, https://doi.org/10.5047/eps.2011.05.034, 2011.

Kang, D. J., Chen, W. K., Zhao, H. Q., and Wang, D.: Rapid Assessment of the September 5, 2022 Ms 6.8 Luding Earthquake in Sichuan, China. Earthquake Research Advances, 100214, https://doi.org/10.1016/j.eqrea, 2023.

Yao, Q., Wang, D., Fang, L. H., and Mori, J.: Rapid estimation of magnitudes of large damaging earthquakes in and around Japan using dense seismic stations in China, Bull. Seismol. Soc. Am. 109, 2545–2555. https://doi.org/10.1785/0120190107, 2019.

Zhao, H. Q., Jia, Y. J., Chen, W. K., Kang, D. J., and Zhang, C.: Rapid mapping of seismic intensity assessment using ground motion data calculated from early aftershocks selected by GIS spatial analysis, Geomatics, Natural Hazards and Risk, 14:1, 1-21. https://doi.org/10.1080/19475705.2022.2160663, 2023.

3. To better validate the accuracy of the source dimension estimated from the early aftershocks, the authors could compare your results with source ruptures, at least for large earthquakes. I believe there are many cases that can be utilized for such comparison.

**Reply:** Eleven earthquakes with Mw ≥7.0 were used as an example in Section 3.2. Our results were compared to surface rupture lengths calculated using an empirical formula for wells and those documented in the literature, and the linear directional mean of surface rupture was calculated using ArcGIS software (Table 3). Subject to the conditions of use, the fitting results of our method can provide reasonably accurate information concerning the length and direction of surface rupture.

In addition, we include a comparison with the back-projection results of Chen et al. (2022a) in this section, which supports our conclusions (lines 363–381). However, as Lowess is essentially a nonparametric regression method that ignores the complex physical relationships contained in the aftershock sequence, we believe its results cannot fully replace those obtained through physical means (e.g., back-projection techniques), but the different methods can be cross-referenced to make further corrections. The added content is as follows:

"We gathered data on the estimated source rapture of the back-projection algorithm for the Wenchuan earthquake (Chen et al., 2022a). Using the same technique, a set of results reflecting the surface rupture of the 2016 Kaikura Mw 7.8 earthquake was calculated using waveform data from high sensitivity seismograph network in Japan. Both the Lowess and back-projection results show rupture directions similar to those indicated by the long axis of the isoseismal line in the area with intensity VIII of the Wenchuan earthquake, but the former estimates a longer rupture length (Fig.11(a)). Furthermore, the back-projection results reveal more details concerning the rupture. For example, the back-projection results indicate a possible fracture near the IX-degree intensity anomaly in the long-axis direction. This method has also demonstrated benefits in determining the intensity anomaly area for the 2022 Maduo Mw7.3 earthquake (Chen et al, 2022b). As a nonparametric method, the points fitted by Lowess are clearly distributed along a curve. However, when the fault system in the seismogenic region is complex, the dominant orientation of the rupture traced using the back-projection method may be problematic (Fig.11(b)). A clear guide to array data selection may be required when using the back-projection method, and we recognize that the results of array data calculations are more accurate when the appropriate region is chosen (Wang and Hutko, 2018). Aftershocks that have been relocated can be used to determine rupture fault trajectories, and their combination with inverse projection techniques has been applied to determine transient shear ruptures (Li et al., 2019; Cheng et al., 2023). These two methods could be cross-referenced in application for more accurate intensity evaluation results overall."

[Figure]

**Figure 11**:Comparison of surface rupture results obtained using the Lowess and back-projection methods for the (a) 2008 Wenchuan Mw 7.9 and (b) 2016 Kaikōura Mw 7.8 earthquakes.

4. Comparison of your results with Chen et al. (2022a, b) that you already cited in this work is also beneficial.

**Reply:** We have added a comparison with Chen et al.'s (2022a, b) work to both the examination of the source rupture results and the discussion of time efficiency, which adds to the richness of our manuscript. The additions are mentioned above in the responses to major comments 2 and 3.

**Minor comments (RC1):**

Line 10, mainshocks

Line 13, of 59 M XXX~XXX earthaukes that occurred from 2000-2022

Line 21, Our study suggest that with early accessible aftershocks, we are able to rapidly determine the rupture fault plane (s), thus have better estimae of the seismic intensities.

Line 44, of an earthquake is limited,

Line 47, after earthquakes

Line 94, We selected Mw $\geqslant$ 6.6 shallow earthquakes that occurred during 2000-2022 in this study.

...

**Reply:** We have checked the language errors in the manuscript and corrected them, polishing the language overall.

**Major comments (RC2):**

**1. I believe the introductory part of the work should be greately improved before being published.**

**Reply:** We appreciate your comments regarding the introduction of our manuscript. In accordance with the comments in the supplement, we made considerable revisions to the introduction section. Almost all of the sentences were rewritten without altering the intended meaning of the original text. In the new version, we have paid special attention to sentence structure, grammar, and the transitions between texts. Several details have been elaborated in light of the additional comments. The updated introduction was expanded from four to five paragraphs, making the content of each paragraph more distinct. The significant modifications are shown below.

(1) Based on the review, we rewrote the first paragraph of the introduction and included references.

**Original text:** "Seismic intensity reflects the strength of ground motion and its influence. Rapid seismic intensity assessment helps in formulating an early emergency response after a destructive earthquake. The rapid and accurate output of seismic intensity assessment could notably reduce the loss of life and property in disaster areas. Therefore, it is necessary to develop methods for the faster assessment of seismic intensity and the efficient use of disaster data in the early post-earthquake period."

**Comments in supplement:**

[Figure]

➢ Adequate references are required!
➢ needs some work on the grammer.

**After modification:** "Seismic intensity reflects the strength of ground motion caused by an earthquake and its influence at a certain location. Rapid and accurate assessment of seismic intensity facilitates the development of emergency measures in the aftermath of a destructive earthquake, thereby reducing the number of fatalities and property damage (Erdik et al., 2011; Poggi et al., 2021). Therefore, it is necessary to develop methods for the rapid assessment of seismic intensity and the effective utilization of disaster data in the early post-earthquake period."

References added:

Erdik, M., Şeşetyan, K., Demircioğlu, M. B., Hancılar, U., and Zülfikar, C.: Rapid earthquake loss assessment after damaging earthquakes. Soil Dyn. Earthq. Eng., 31(2), 247-266. https://doi.org/10.1016/j.soildyn.2010.03.009, 2011

Poggi, V., Scaini, C., Moratto, L., Peressi, G., Comelli, P., Bragato, P. L., and Parolai, S.: Rapid damage scenario assessment for earthquake emergency management. Seismol Res Lett., 92(4), 2513-2530. https://doi.org/10.1785/0220200245, 2021.

(2) Lines 33-35 of the original text have had their content optimized.

**Original text:** "ShakeMap, one of the world's established platforms for distributing seismic information, utilises a combination of recorded and estimated values of ground motion to assess the seismic intensity in a region (Worden et al., 2020)."

**Comments in supplement:**
➢ ShakeMap: Reference, webpage?

**After modification:** "The ShakeMap system of the US Geological Survey (USGS) combines predicted ground motion values with station observations to determine the seismic intensity of a region and publishes the results online in near real-time (Worden et al., 2020)."

(3) Lines 41–44 of the original text were rewritten, and the paragraph was split into two.

**Original text:** "From the perspective of data acquisition, the time from the occurrence of the earthquake to the first acquisition of disaster data from the disaster area (generally within a few hours after the mainshock) is considered as the black box period of earthquake emergency disaster service."

**Comments in supplement:**
➢ can you please re-write this? Do you have any reference for this?

**After modification:** "The time between the occurrence of an earthquake and the first acquisition of disaster data from the seismogenic region, typically within 2–3 hours of the mainshock, is defined as the black box period for earthquake emergency response (Nie and An, 2013)."

Reference added:
Nie G, and An J.: Basic theoretical model of earthquake emergency response (in Chinese). Urban Disaster Reduct. 3:25–29. 2013.

(4) Lines 50–57 have had their content optimized.

**Original text:** "To expand the method of rapid seismic intensity assessment and improve its timeliness and accuracy, Chen et al. (2022a) proposed a method to predict the source rupture process by using the far-field seismic array data back-projection technique, and combining it with the ground motion prediction equation (GMPE) for rapid assessment of seismic intensity. This method was validated in the 2021 Maduo Mw 7.3 earthquake in Qinghai province and the Yangbi Mw 6.1 earthquake in Yunnan province (Chen et al., 2022b). However, accurate inversion of the source rupture process for earthquakes that occur in different regions and selection of more applicable regional GMPEs are the key points that still need to be addressed and improved in the Chen et al. method."

**After modification:** "Back projection could image the fault geometry of large earthquakes at high resolution and is frequently used to trace surface rupture processes and source durations (Ishii et al., 2005; Wan et al., 2022). The combination of back-projection results and P-wave amplitudes could be used to quickly estimate the source length and magnitude of large earthquakes (Wang et al., 2017). Using the back-projection technique and ground motion prediction equation (GMPE), Chen et al. (2022a) developed a new algorithm for quickly obtaining the intensity maps of destructive earthquakes. The algorithm was validated during the emergency response phase of the 2021 Maduo Mw 7.3 and 2021 Yangbi Mw 6.1 earthquakes in China and was confirmed to be suitable for intensity assessment in regions with sparse observation networks (Chen et al., 2022b)."

References added:

[revised manuscript text omitted]

Moreover, language issues present in other sections of the manuscript have been reviewed and corrected.

**2. In addition, arrangement and presentation of tables and figures for chosen earthquakes needs to be enhanced.**

**Reply:** All figures in the manuscript were re-exported, with errors in the figures corrected and content enriched. Furthermore, we have rearranged the figures in the manuscript and added two figures for the spatiotemporal distribution of aftershocks and the results of the intensity assessment of the two earthquakes that occurred in Turkey in 2023, which we will introduce in the response to major comment 3. Considering that the presence or absence of Figs. 7 and 12 in the original manuscript had less impact on the its content and the acquisition of the pertinent conclusions, we removed these two images and improved the description of the others, streamlining the content of the relevant sections to make what must be expressed clearer. In conjunction with the comments in the supplementary document, we have modified the presentation of Fig. 11, and the new image demonstrates the effect of the AL-SM99 method application more clearly. In response to the comments in the supplementary file, we added a table containing the results of outlier checks for significant cases in the manuscript. Data for two earthquakes that occurred in Turkey in 2023 were added to Tables 3 and 4. The modified contents are shown below.

(1) Lowess results for the two 2023 Turkey earthquakes are added in Figure 9. The graphical captions now include literature sources for actual surface rupture data.

**Original figure:**

[Figure]

**Figure 10:** Comparison of Lowess-fitted curves with actual surface rupture for the (a) 2005 Kashmir Mw 7.6; (b) 2008 Wenchuan Mw 7.9; (c) 2010 Baja California Mw 7.2; (d) 2011 Van Mw 7.1; (e) 2016 Kumamoto Mw 7.0; (f) 2016 Kaikōura Mw 7.8; (g) 2018 Palu Mw 7.5; (h) 2019 Ridgecrest Mw 7.1; and (i) 2021 Maduo Mw 7.3 earthquakes.

**After modification:**

[Figure]

**Figure 10:** Comparison of Lowess-fitted curves with actual surface rupture for the (a) 2005 Kashmir Mw 7.6; (b) 2008 Wenchuan Mw 7.9; (c) 2010 Baja California Mw 7.2; (d) 2011 Van Mw 7.1; (e) 2016 Kumamoto Mw 7.0; (f) 2016 Kaikōura Mw 7.8; (g) 2018 Palu Mw 7.5; (h) 2019 Ridgecrest Mw 7.1; (i) 2021 Maduo Mw 7.3 (j) 2023 Pazarcık Mw 7.8 and (k) 2023 Elbistan Mw 7.5 earthquakes. Fault rupture traces were extracted from relevant literature or downloaded from ShakeMap, then digitised in ArcGIS (Kaneda et al., 2008; Li et al., 2008; Fletcher et al., 2014; Liu et al., 2015; Toda et al., 2016; Shi et al., 2019; Zhang et al., 2021; Reitman et al., 2023).

(2) Modification of Figure 11.

**Comments in supplement:**

➤ Can you enhance the presentation quality of this figure please?

**Original figure:**

[Figure]

**Figure 11:** Average residual curves of PGA and PGV for 24 earthquakes, which have station-recorded. (a) Mean residuals of PGV and (b) mean residuals of PGA. The average residuals calculated in steps of 10 km within 200 km from the epicentre are connected into curves.

**After modification:** "The model prediction results are credible if the aftershocks accurately reflect the information of the causative faults as the input data of SM99 GMPE.

The average residuals of the PGVvs30 and PGA predicted values for the 23 earthquakes were between -0.4 and 0.4 (Fig. 13). With increasing magnitude, the residuals of ground motion prediction decrease significantly. The residuals of ground-motion predictions for earthquakes with magnitudes of 7.5–8.3 are superior to those of the other two subgroups, whereas the residuals of Mw 6.0–6.5 are higher. This implies that the method is more applicable in large-magnitude earthquakes. For many earthquakes shown in Fig. 13, the residuals of the ground motion prediction results increase with distance, indicating the advantage in determining the extent of the hardest hit areas."

[Figure]

**Figure 13:** Heat map of the average residuals of predicted (a) PGA and (b) PGVvs30 values for 23 earthquakes, which have good station records. A residual value is calculated for every 10 km increase in the range of 200 km from the epicentre, and the corresponding colour is assigned to the corresponding position in the graph. The magnitudes were divided into three groups. Each row represents an earthquake, and the histogram on the left displays the associated magnitude.

(3) We optimized Fig. 14 by reducing the number of elements in Fig. 14(a) and representing the fitted curves for aftershocks over different time periods with gradient colours. Additionally, the units of the data for the PGVvs30 grading interval in the legend of Fig. 14(b) were changed from m/s to cm/s, which is consistent with the illustration of the full-text intensity assessment results.

**Original figure:**

[Figure]

**Figure 14:** The 2016 Kaikōura Mw 7.8 earthquake's Lowess split-time fitting results. (a) Lowess fitting curves plotted at 0.5 h intervals; and (b) seismic intensity map assessed based on 1.5 h aftershocks fitting result.

**After modification:**

[Figure]

**Figure 15:** Lowess split-time fitting results for the 2016 Kaikōura Mw 7.8 earthquake. (a) Lowess fitting curves plotted at 0.5 h intervals; and (b) assessment of seismic intensity using aftershocks within 1.5 hours of the earthquake.

(4) Figure 7 in the original manuscript has been removed. The explanation of Fig. 7 does not highlight the benefits of AL-SM99 because we used too much text in Section 3.1.2 to describe how this figure was drawn. The seismic intensity assessment of the results of the Kaikura earthquake, in contrast, is poorly described. Therefore, we have revised this section to emphasize the effectiveness and benefits of AL-SM99 for seismic intensity assessment.

**Removed illustration:**

[Figure]

**Figure 7:** Residuals and average residuals of PGV prediction results; station observations were downloaded from ShakeMap. (a) Average residuals within 100 km and (b) average residuals within 200 km.

(5) Figure 12 in the original manuscript has been removed. In addition to the earthquake cases discussed in the study, we have applied the AL-SM99 to more earthquakes worldwide. In the majority of seismic cases, however, monitoring station data are scarce. In the original manuscript, we wanted to demonstrate the accuracy of our method by comparing the area of the hardest-hit area assessed by AL-SM99 to the area of the hardest-hit area assessed by ShakeMap (or CEA), that is, the ratio of the overlap hardest-hit area to that of the other methods. However, the figure only partially illustrates the previously elucidated conclusions, such as the reliability of AL-SM99 and the regional restrictiveness of GMPE, and it does not present any new findings. Instead, it makes this section a lengthy textual presentation. Therefore, we have removed this illustration and simplified this section.

[Figure]

**Figure 12:** Violin plot of the ratio of the area of the hardest-hit areas assessed in this study to that of the hardest-hit areas assessed by ShakeMap for 59 earthquakes.

(6) Added a new table.

**Comments in supplement:**

➢ Can you put all the events in a table with their details and number of total and deleted aftershocks and refer the reader to that table when required?

**After modification:** The revised manuscript includes a new table that tallies the number of aftershocks eliminated from the examples and provides additional information (location, magnitude, etc.) for these cases.

**Table 1:** Number of aftershocks and identified outliers for the selected earthquakes.

| | Data | Location | Magnitude | Aftershocks | Outliers |
|---|---|---|---|---|---|
| 1 | 20001006 | Matsue (Japan) | 6.7 | 152 | 4 |
| 2 | 20030526 | Miyagi-Oki (Japan) | 7.0 | 259 | 24 |
| 3 | 20051008 | Kashmir (Pakistan) | 7.6 | 54 | 0 |
| 4 | 20080512 | Wenchuan (China) | 7.9 | 43 | 0 |
| 5 | 20080613 | Iwate-Miyagi Nairiku (Japan) | 6.9 | 227 | 1 |
| 6 | 20100404 | Baja California (Mexico) | 7.2 | 60 | 2 |
| 7 | 20100903 | Darfield (New Zealand) | 7.0 | 139 | 2 |
| 8 | 20110411 | Hamadoori (Japan) | 6.6 | 79 | 12 |
| 9 | 20111023 | Van (Turkey) | 7.1 | 46 | 6 |
| 10 | 20130816 | Grassmere (New Zealand) | 6.5 | 46 | 3 |
| 11 | 20150425 | Gorkha (Nepal) | 7.8 | 68 | 14 |
| 12 | 20150916 | Illapel (Chile) | 8.3 | 56 | 4 |
| 13 | 20160415 | Kumamoto (Japan) | 7.0 | 538 | 0 |
| 14 | 20161030 | Preci (Italy) | 6.6 | 89 | 6 |
| 15 | 20161113 | Kaikōura (New Zealand) | 7.8 | 106 | 0 |
| 16 | 20171112 | Sarpol-e Zahab (Iraq) | 7.4 | 15 | 2 |
| 17 | 20180504 | Hawaii (America) | 6.9 | 38 | 1 |
| 18 | 20180905 | Tomakomai (Japan) | 6.6 | 162 | 6 |

| 19 | 20180928 | Palu (Indonesia) | 7.2 | 18 | 2 |
|----|----------|------------------|-----|-----|---|
| 20 | 20181130 | Anchorage (America) | 7.1 | 127 | 9 |
| 21 | 20190706 | Ridgecrest (America) | 7.0 | 105 | 2 |
| 22 | 20201030 | Samos (Greece) | 7.0 | 97 | 10 |
| 23 | 20210521 | Maduo (China) | 7.3 | 70 | 1 |
| 24 | 20220107 | Menyuan (China) | 6.6 | 43 | 4 |
| 25 | 20220905 | Luding (China) | 6.6 | 78 | 8 |
| 26 | 20230206 | Pazarcik (Turkry) | 7.8 | 27 | 5 |
| 27 | 20230206 | Elbistan (Turkey) | 7.5 | 24 | 0 |

**3. I was not entirely convinced if the propsed technique was efficient but I hope re-writing the results could clear up the benefits.**

**Reply:** The result section has been rewritten, and the interpretation of the graphical and tabular content has been improved. The rewritten conclusions highlight two advantages of AL-SM99: reasonable judgment of rupture pattern and direction in simple and well-defined fault systems in the seismogenic region, and reliable indication of overall rupture direction and rupture length in complex fault systems. In the results section, we have added a comparison between the rupture results of the physical means inversion and the Lowess fit result. The comparison also demonstrates that the seismic intensities estimated by the method are reasonable under the conditions of use.

Notably, there are conditions for the use of this method, and the Lowess is a nonparametric regression method that ignores the complex physical relationships contained in the aftershock sequence. Therefore, we believe its results cannot fully replace those obtained through physical means (e.g., back-projection techniques). However, the different methods can be cross-referenced to make further corrections to the results.

We tested our method for the 2022 Luding Mw 6.6 earthquake in China (Kang et al., 2023), as well as two great earthquakes (Mw 7.8 and Mw 7.5) in Turkey in 2023. The latter have been added to the results section of the revised manuscript. The results show that this method is feasible in the given conditions. Furthermore, we improved the method proposed in this study such that it could be used to assess the seismic intensity of small magnitude earthquakes as well (Zhao et al., 2023). Our recent publications that contain related works are listed below.

[revised manuscript text omitted]

Map of intensity assessed with AL-SM99 on the day of the earthquake. The time displayed is Beijing time.

[Figure]

[Figure]

Seismic intensity results of Turkey Mw 7.8 (version 15) and Mw 7.5 (version 9) earthquakes released by USGS are as follows.
(https://earthquake.usgs.gov/earthquakes/eventpage/us6000jllz/shakemap/intensity; https://earthquake.usgs.gov/earthquakes/eventpage/us6000jlqa/shakemap/intensity, last access: 15 March 2023).

[Figure]

Macroseismic Intensity Map USGS
ShakeMap: 5 km SSE of Ekinözü, Kahramanmaraş, TR
Feb 06, 2023 10:24:50 UTC M7.5 N38.02 E37.21 Depth: 15.0km ID:us6000jlqa

| SHAKING | Not felt | Weak | Light | Moderate | Strong | Very strong | Severe | Violent | Extreme |
|---|---|---|---|---|---|---|---|---|---|
| DAMAGE | None | None | None | Very light | Light | Moderate | Moderate/heavy | Heavy | Very heavy |
| PGA(%g) | <0.0464 | 0.297 | 2.76 | 6.2 | 11.5 | 21.5 | 40.1 | 74.7 | >139 |
| PGV(cm/s) | <0.0215 | 0.135 | 1.41 | 4.65 | 9.64 | 20 | 41.4 | 85.8 | >178 |
| INTENSITY | I | II-III | IV | V | VI | VII | VIII | IX | X+ |

Scale based on Worden et al. (2012)          Version 9: Processed 2023-03-08T17:21:31Z
△ Seismic Instrument  ○ Reported Intensity       ★ Epicenter  ▭ Rupture

earthquake was calculated using waveform data from high sensitivity seismograph network in Japan. Both the Lowess and back-projection results show rupture directions similar to those indicated by the long axis of the isoseismal line in the area with intensity VIII of the Wenchuan earthquake, but the former estimates a longer rupture length (Fig.11(a)). Furthermore, the back-projection results reveal more details concerning the rupture. For example, the back-projection results indicate a possible fracture near the IX-degree intensity anomaly in the long-axis direction. This method has also demonstrated benefits in determining the intensity anomaly area for the 2022 Maduo Mw7.3 earthquake (Chen et al, 2022b). As a nonparametric method, the points fitted by Lowess are clearly distributed along a curve. However, when the fault system in the seismogenic region is complex, the dominant orientation of the rupture traced using the back-projection method may be problematic (Fig.11(b)). A clear guide to array data selection may be required when using the back-projection method, and we recognize that the results of array data calculations are more accurate when the appropriate region is chosen (Wang and Hutko, 2018). Aftershocks that have been relocated can be used to determine rupture fault trajectories, and their combination with inverse projection techniques has been applied to determine transient shear ruptures (Li et al., 2019; Cheng et al., 2023). These two methods could be cross-referenced in application for more accurate intensity evaluation results overall.

[Figure]

**Figure 11**:Comparison of surface rupture results obtained using the Lowess and back-projection methods for the (a) 2008 Wenchuan Mw 7.9 and (b) 2016 Kaikōura Mw 7.8 earthquakes.

**4. Scientifically, I think a physics-based simulation would be an approperaite way of proving the point and it would not be an entirly hard task to do.**

**Reply:** The 2021 *Journal of Geophysical Research: Solid Earth* article on the simulation of the relationship between earthquake sequences and geometrically complex faults (Ozawa and Ando, 2021) and the 2017 *Earth, Planets, and Space* article on the relationship research between mainshock ruptures and aftershock sequences based on dense seismic observations (Yukutake and Iio, 2017) served as a source of inspiration for us. The goal of this study was to broaden the use of early aftershock data and serve as a reference for testing the accuracy of the method proposed earlier by our team (Chen et al., 2022). This method can be combined with energy point data obtained via the inverse projection algorithm to screen energy points and visualize fault rupture trends. We will focus on using physical

simulations to validate our approach in the next step of our work. The method will also be compared to rupture estimates obtained from remote sensing, finite tomography, and inverse projection techniques.

The acronym does not match the statement.

**Reply:** Robust locally weighted regression (Cleveland, 1979) is a method for smoothing scatterplots in which the fitted value at Xk is the value of a line fit to the data using weighted least squares where the weight for (xi,yi) is large if xi is close to Xk and small if xi is not close to Xk. Cleveland (1981) published an article called "LOWESS: A Program for Smoothing Scatterplots by Robust Locally Weighted Regression" and gave a brief introduction to the program. Later, in 1988, he published "Locally Weighted Regression: An Approach to Regression Analysis by Local Fitting". He called this method Loess and explained it in this article. "Locally weighted regression, or loess, is a way of estimating a regression surface through a multivariate smoothing procedure, fitting a function of the independent variables locally and in a moving fashion analogous to how a moving average is computed for a time series; it is a straightforward extension of the univariate loess smoother discussed by Cleveland (1979) (Cleveland and Devlin, 1988)." To distinguish it from Loess, Lowess is frequently used to describe locally weighted regressions in univariate scenarios, i.e., the original method proposed by Cleveland in 1979 (Mariani and Basu, 2014). Of course, certain sources consider the two terms to be different names for the same method (https://www.itl.nist.gov/div898/handbook/pmd/section1/pmd144.htm).
The R language software distinguishes between these two terms as well. The differences between the methods in R software depicted under the two names can be found in the forum discussion (https://support.bioconductor.org/p/2323/). The name of the current manuscript, which was chosen after careful consideration, is relatively succinct and descriptive of our research. The term Lowess in parentheses is not a straightforward abbreviation of the preceding phrase, but

rather the original name of the program it describes. Moreover, Lowess is used to differentiate it from Loess in order to reflect our use of the method more accurately.

"…In this study, only the coordinate position of the aftershock is utilised when fitting the aftershock sequence with Lowess. There is still a gap between the curve length and local trend obtained by fitting and the actual surface rupture. In future work, the type of the causative fault and the geological environment of the seismogenic area can be taken into consideration, and the empirical formula such as Wells' surface rupture formula can be utilised to correction. In addition, it is useful to study the aftershock sequence relocation method to improve the fitting accuracy, and take an in-depth look into the relationship between the spatial distribution and genesis of early

aftershock sequences and the causative fault. Lowess is also worth discussing with regard to the application of smoothing the spatial distribution trend of aftershocks over a long period, and the possibility exists to combine aftershock predictions to achieve seismic intensity prediction."

**Comment:** This would be very interesting. I think you could mention a few references here.

**After modification:**

"In this study, we developed a method for evaluating seismic intensities based on aftershock data gathered within 2 hours of the mainshock. Aftershock sequences are treated as scatterplots, with Lowess fitting applied to their longitude and latitude coordinate values. The result of the fit is used to roughly describe the fault rupture trend, and the SM99 GMPE was used to calculate ground motion data. The PGV values were then converted to seismic intensity. The main conclusions are as follows:

1.    The length and direction of the surface rupture can be roughly outlined by the early aftershock sequence following the mainshock. The fitted curves from Lowess are helpful for pinpointing the location of causative faults and rupture scales. When the fault system in the seismic region is clear and simple, the Lowess fitted curves can be used to accurately determine the location and length of the fault rupture. When the fault system is complex, Lowess results can still indicate the overall rupture trend and make reliable rupture scale judgments.

2.    Lowess is suited for aftershock sequences of large magnitude earthquakes (Mw $\geq$ 7.0). The fitted curves are always slightly longer than the actual surface rupture, indicating that aftershocks occurred at a certain distance from the tips of the fault shortly after the mainshock (Ozawa and Ando, 2021). This method broadens the scope of application for early post-earthquake aftershock data.

3.    Aftershocks frequently cause secondary damage to buildings in the affected region, resulting in greater economic losses or fatalities. The seismic intensity map based on the spatial distribution trend assessment of aftershock sequences could reflect the extent of the hardest-hit areas and regions where cause property damage and fatalities may occur.

4.    When the listed conditions are met, the seismic intensities assessed using AL-SM99 can serve as a useful reference for early earthquake emergency response efforts. The outcomes of intensity assessment may also provide a basis for different perspectives in studying the radiative energy of earthquakes and locating causative faults. Obviously, selecting the appropriate GMPEs can produce more accurate intensity assessment results.

Notably, only the coordinate positions of the aftershocks are used when fitting the aftershock sequence with Lowess. A discrepancy remains between the fitted curve length, local trend, and the actual surface rupture. In future research, the type of the causative faults and geological context of the seismogenic regions will be considered, and empirical correction formulas such as Wells' surface rupture formula will be used for correction. It is beneficial to study the aftershock sequence relocation methods and the relationship between the spatial distribution of early aftershock sequences and causative faults. The application of Lowess to smoothing the spatial distribution trends of aftershock sequences over extended time periods is also of interest. AL-SM99 can dynamically generate intensity assessment results in conjunction with aftershock monitoring networks. Although the viability of aftershock prediction remains debatable, it is possible to combine aftershock predictions and

achieve rapid seismic intensity prediction (DeVries, et al., 2018; Mignan and Broccardo, 2019)."

References added:

DeVries, P. M., Viégas, F., Wattenberg, M., and Meade, B. J.: Deep learning of aftershock patterns following large earthquakes. Nature, 560(7720), 632-634. https://doi.org/10.1038/s41586-018-0438-y, 2018.

Mignan, A., and Broccardo, M.: One neuron versus deep learning in aftershock prediction. Nature 574: E1–E3. https://doi.org/10.1038/s41586-019-1582-8, 2019.

**Other Modification Notes:**

(1) We have rewritten the abstract to clarify the advantages and benefits of the new method.

**After modification:**

Accurate and rapid assessment of seismic intensity after a destructive earthquake is essential for efficient early emergency response. We proposed an improved method, AL-SM99, to assess seismic intensity by analysing aftershock sequences that occur within 2 hours of mainshocks. The implementation effect and application conditions of this method were illustrated using 27 earthquakes with Mw 6.5–8.3 that occurred globally between 2000 and 2023. When the fault system in the seismic region is clear and simple, the robust locally weighted regression program (Lowess)-fitted curves could be used to estimate the location and length of the fault rupture. Lowess results can indicate the overall rupture trend and make reliable rupture scale judgments even when the fault system is complex. When Mw $\geq$ 7.0 and the number of aftershocks exceeds 40, the AL-SM99 intensity evaluation results may be more reliable. Using aftershock catalogues obtained by conventional means allows for a stable assessment of seismic intensities within 1.5 hours of the mainshock. When the number of aftershocks is sufficiently large, the intensity assessment time can be greatly reduced. With early accessible aftershocks, we can quickly determine the rupture fault planes and have a better estimate of the seismic intensities. The results of the intensity assessment provide a useful guide for determining the extent of the hardest-hit areas. By expanding the data sources for seismic intensity assessment, the early accessible data are utilised adequately. This study provides a valuable reference point for investigating the relationship between early aftershock events and fault rupture.

(2) In Section 2.1, a description of earthquake damage and human perception for intensity VII has been added.

**After modification:**

When the Modified Mercalli Intensity (MMI) is VII, ShakeMap uses the terms "very strong" and "moderate damage" to describe the levels of impact on a region (Worden et al., 2020). Similar descriptions of intensity VII exist in the Chinese Seismic Intensity (CSI) scale. For the intensity range of VII–VIII, human perception of shaking began to saturate, and it may be difficult to distinguish seismic intensities above VII based on the individual descriptions of the felt shaking alone. (Dengler and Dewey, 1998; Worden et al., 2020).

This research was supported by the Major Science and Technology Projects of Gansu Province (21ZD4FA011) and the National Key Research and Development Program of China (No. 2017YFB0504104).